# Break the Block: Dynamic-size Reasoning Blocks for Diffusion Large Language Models via Monotonic Entropy Descent with Reinforcement Learning

Yan Jiang [1]    Ruihong Qiu [1]    Zi Huang [1]

## Abstract

Recent diffusion large language models (dLLMs) have demonstrated both effectiveness and efficiency in reasoning via a block-based semi-autoregressive generation paradigm. Despite their progress, the fixed-size block generations remain a critical bottleneck for effective and coherent reasoning. (i) From a global perspective, **different reasoning tasks would correspond to different optimal decoding block sizes**, which makes a "one-size-fits-all" assumption ineffective. (ii) Even **within a single reasoning task, the rigid block partitioning would break the logical flow and reduce reasoning coherence.** Through empirical observations, we reveal that for block-wise entropy, incorrect reasoning exhibits a fluctuating and unsteady trend between blocks, whereas the correctly generated tasks follow a consistent descending trend. Therefore, this paper proposes *b1*, a novel post-training framework for dLLMs that learns dynamic-size reasoning blocks via a **Monotonic Entropy Descent** objective with reinforcement learning to enhance reasoning coherence. *b1* integrates seamlessly as a plug-and-play module with existing dLLM's post-training algorithms. Extensive experiments across various reasoning benchmarks showcase *b1*'s consistent improvement over existing fixed-size block baselines. Our code has been released at https://github.com/YanJiangJerry/Block-R1.

## 1. Introduction

Diffusion large language models (dLLMs) (Zhu et al., 2025a; Ye et al., 2025; Arriola et al., 2025) have emerged as compelling alternatives to autoregressive (AR) models. Unlike the token-by-token causal generation in AR frame-

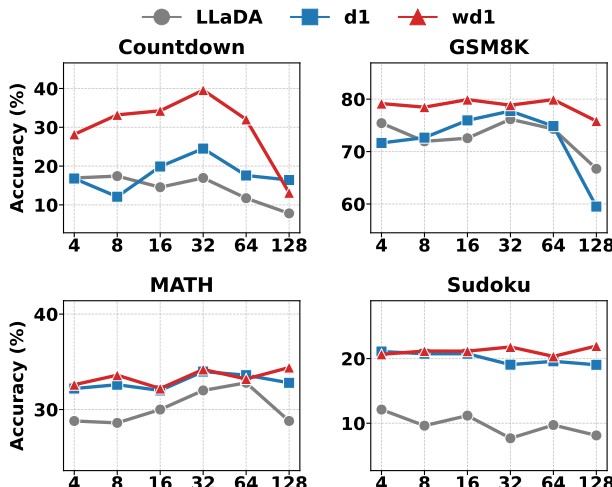

*Figure 1.* **Analysis of block size against reasoning performance in dLLMs.** The x-axis indicates block size for generation. Results demonstrate that the optimal block size varies significantly across different reasoning benchmarks.

works, dLLMs generate tokens in a parallel manner, mostly within blocks (Nie et al., 2025; Zhu et al., 2025a). Specifically, a sequence is separated into multiple blocks with a pre-defined fixed size, whereby parallel token generation is conducted within these individual blocks. Such block-based generation (Nie et al., 2025; Zhu et al., 2025a; Arriola et al., 2025) demonstrates substantial potential in real-world applications like mathematical reasoning (Xiong et al., 2025).

Despite previous progress, the inherent fixed-size block generation in dLLMs remains a significant challenge for complex reasoning tasks. (i) On the one hand, the optimal block size for different datasets is different, as demonstrated in Figure 1. Nonetheless, existing dLLMs mostly select a fixed size for block generation across various tasks. **Such a "one-size-fits-all" constraint may limit dLLMs' generation potential**, especially in reasoning where the reasoning trace of a model would largely impact the outcomes. (ii) On the other hand, even within the generation process for a single input prompt, **enforcing rigid block partition would severely disrupt the logical flow of the reasoning trace, undermining coherence and ultimately resulting in the generation of incorrect answers.**

Given these challenges, a critical question arises:

[1] School of Electrical Engineering and Computer Science, The University of Queensland, Brisbane, Queensland, Australia . Correspondence to: Yan Jiang <yan.jiang@uq.edu.au>.

*Proceedings of the 43rd International Conference on Machine Learning*, Seoul, South Korea. PMLR 306, 2026. Copyright 2026 by the author(s).

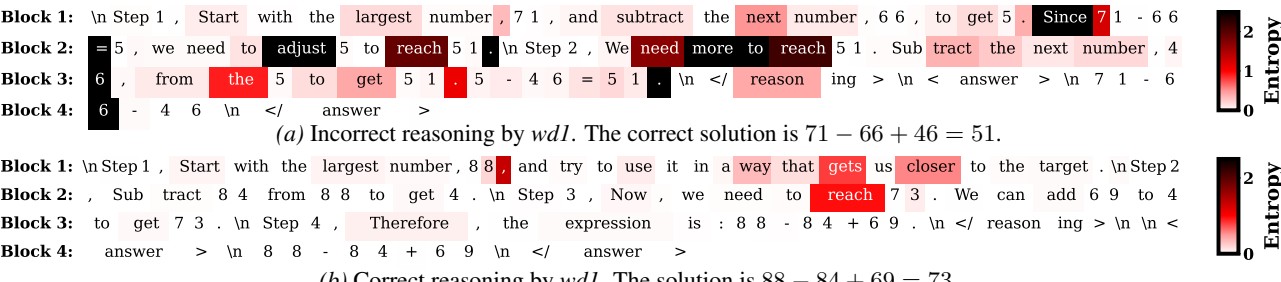

*(a)* Incorrect reasoning by *wd1*. The correct solution is $71 - 66 + 46 = 51$.

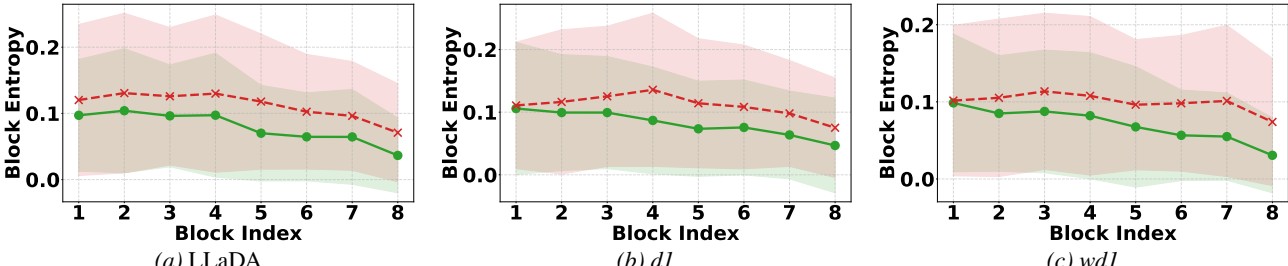

*(b)* Correct reasoning by *wd1*. The solution is $88 - 84 + 69 = 73$.

*Figure 2.* **Token-wise entropy comparison between incorrect and correct reasoning** with fixed-size blocks by *wd1* on the Countdown reasoning benchmark. (a) Rigid boundaries disrupt numerical operations (e.g., splitting "$71 - 66$" between Block 3 and 4), causing high-entropy anomalies and incorrect results. (b) Coherent reasoning is maintained when block boundaries do not interrupt calculations, reflected by a low and descending entropy pattern across the reasoning generation.

*(a)* LLaDA     *(b)* d1     *(c)* wd1

*Figure 3.* **Block-wise entropy evolution for LLaDA, *d1*, and *wd1*** by their default block size of 32 on Math500. Red and green lines denote block generations for incorrect and correct reasoning results, respectively. Shaded areas represent standard deviation. The block entropy is calculated as the mean token-wise entropy within the same block. While correct reasoning generations exhibit a consistent descent in block entropy, the incorrect ones fluctuate more significantly.

***Why do fixed-size blocks hinder the reasoning coherence?***

Our observations indicate that dLLMs' reasoning coherence is significantly compromised by the rigid partitioning inherent to fixed-size block generation, reflected by the entropy of tokens and blocks. Specifically, through empirical analysis on **token-level** entropy in Figure 2, we observe that for the incorrect example, the fixed boundary rigidly separate the reasoning trace in an unexpected place for the logic, e.g., a block boundary located between "$71 - 6$" and "$6 - 46$", which leads to a high entropy pattern for the generated tokens. While for the correctly predicted example, a proper cut-off without interrupting the calculation would not impact the reasoning coherence between blocks, reflecting a relatively low entropy pattern with a descending entropy trend throughout the reasoning trace. Moreover, from a **block-level** view in Figure 3 by averaging token entropy within blocks, the fixed-size block creates a disparity in block entropy during reasoning for correct and incorrect tasks. Specifically, the reasoning traces with correct results demonstrate a monotonic descending trend in block-wise entropy (green), whereas incorrect traces are plagued by fluctuating or stagnant entropy (red) across blocks. Such observations suggest that, for an effective reasoning task, the model confidence for each generated block should progressively improve along the reasoning process. In contrast, if the reasoning is incorrect, the model often hesitates and demonstrates a state of high uncertainty in block entropy.

In light of the above discussion, this paper proposes a novel **dynamic-size reasoning block learning method, termed *b1*.** To achieve dynamic-size reasoning blocks, *b1* learns a block ending indicator with Reinforcement Learning (RL) to identify reasoning step boundaries and semantic endpoints. To enhance reasoning coherence, *b1* introduces **Monotonic Entropy Descent (MED)** via a block entropy reward, facilitating block generations with consistent descending entropy. *b1* represents the first RL framework to break fixed-block constraints, effectively aligning dLLM post-training with its inherent block-based inference. In contrast to existing RL-based methods, *b1* functions as a versatile, plug-and-play module capable of being jointly trained with a wide range of existing RL algorithms. Extensive experiments demonstrate the effectiveness of *b1*. Our main contributions are:

- **Dynamic Reasoning Blocks:** *b1* is the first RL-driven framework for dLLMs to break the rigid, fixed-size block constraints, enabling dynamic-size blocks that aligns with intrinsic reasoning steps for coherent reasoning.
- **Monotonic Entropy Descent:** Empirical observations indicate that block monotonic entropy descent is essential for reasoning. Consequently, a novel RL-based MED objective is developed to enable dLLMs to learn dynamic-size reasoning blocks for coherent and accurate reasoning.
- **Theoretical Insights:** Theoretical analysis establishes that our employed block entropy reward is equivalent to maximising the negative Spearman's rank correlation coefficient, ensuring strictly monotonic entropy descent

across the generated reasoning blocks.
- **Plug-and-Play:** *b1* is seamlessly plug-and-play and consistently improve existing dLLM-based RL frameworks.

## 2. Preliminaries

Let $\mathbf{x} = [x_1, x_2, \ldots, x_L]$ denote a sequence of tokens where $L$ is the sequence length. For dLLMs' block-based generation, the sequence is partitioned into $K$ non-overlapping blocks $\mathcal{B} = \{\mathbf{b}_1, \mathbf{b}_2, \ldots, \mathbf{b}_K\}$ for parallel generations. The index $k$ (where $1 \leq k \leq K$) identifies the $k$-th block and $|\mathbf{b}_k|$ denotes the size of that specific block. In conventional dLLMs' default fixed-size block generation, a constant $c$ enforces a uniform block size $|\mathbf{b}_k| = c$ for all blocks.

### 2.1. Diffusion Large Language Models

**Training.** Diffusion Large Language Models (dLLMs), parametrised by $\theta$, operate by reversing a corruption process where noise is injected into the input by fully replacing the original text sequence with a special [MASK] token. The training objective can be formulated as a denoising cross-entropy loss over a continuous time horizon $t \in [0, 1]$ where $t = 1$ represents the fully masked sequence:

$$\mathcal{L}(\theta) = -\mathbb{E}_{t \sim \mathcal{U}[0,1], \mathbf{x}_t} \left[ \frac{1}{t} \sum_{i=1}^{L} \mathbb{I}(x_t^i = \mathsf{mask}) \log \pi_\theta(x_0^i \mid \mathbf{x}_t) \right], \tag{1}$$

Here $t \sim \mathcal{U}[0, 1]$ represents continuous diffusion time and $\mathbf{x}_t$ denotes the sequence at time $t$. The term $\frac{1}{t}$ serves as a weighting factor for the summation over $L$ tokens. The indicator function $\mathbb{I}(\cdot)$ restricts the loss to masked tokens while $\pi_\theta(x_0^i \mid \mathbf{x}_t)$ refers to the probability predicted by the dLLM for the original token $x_0^i$.

**Block-based Inference.** Unlike AR models that generate tokens strictly one by one, most dLLMs adopt a semi-autoregressive paradigm. Within this framework, a fully masked $L$-length sequence $\mathbf{x}$ is partitioned into non-overlapping blocks $\mathcal{B}$. For standard fixed-size generation, each block $\mathbf{b}_k$ is determined by a constant block size $c$:

$$\mathbf{b}_k = \{i \mid (k-1)c + 1 \leq i \leq \min(kc, L)\} \tag{2}$$

The model generates these blocks sequentially from $k = 1$ to $K$. For each block $\mathbf{b}_k$, the dLLM performs $T$ diffusion steps to denoise all masked tokens $x_i \in \mathbf{b}_k$ in parallel. The block size $c$ directly dictates the degree of parallelism within each denoising step; a larger $c$ enables the simultaneous denoising of more tokens, whereas a smaller $c$ restricts parallel generation to fewer tokens. Consequently, the block size $c$ is the decisive factor in determining the granularity of dLLMs' parallel generation.

### 2.2. Reinforcement Learning for dLLMs

A few recent studies have attempted to apply RL, particularly Group Relative Policy Optimization (GRPO), as a post-training technique to enhance the performance of dLLMs on reasoning tasks (Zhao et al., 2025; Tang et al., 2025; Rojas et al., 2025). For instance, *d1* (Zhao et al., 2025) and *wd1* (Tang et al., 2025) formulate the diffusion-based GRPO objective as:

$$\mathcal{J}_{\text{Diffu-GRPO}} = \mathbb{E}_{\substack{q \sim \mathcal{D}, \\ o_{1:G} \sim \pi_{\text{old}}(\cdot \mid q)}} \left[ \frac{1}{GL} \sum_{g=1}^{G} \sum_{i=1}^{L} \min \left( r_g^i(\theta) \hat{A}_g, \right. \right.$$

$$\left. \left. \text{clip}\left(r_g^i(\theta), 1 - \epsilon, 1 + \epsilon\right) \hat{A}_g \right) - \beta D_{\text{KL}}\left(\pi_\theta \parallel \pi_{\text{ref}}\right) \right] \tag{3}$$

where $r_g^i(\theta)$ denotes the policy ratio $\pi_\theta(o_g^i)/\pi_{\text{old}}(o_g^i)$ for the $i$-th token in the $g$-th completion. In this context, $\pi_\theta$ represents the current policy being optimised while $\pi_{\text{old}}$ signifies the old policy used to generate the group of completions. $\hat{A}_g$ represents the advantage of the $g$-th sequence normalised against the group average of $G$ completions. The hyper-parameter $\epsilon$ bounds the policy update while $\beta$ controls the strength of the KL divergence penalty with respect to the reference policy $\pi_{\text{ref}}$. Unlike exact likelihood computation in AR models, the policy ratio for dLLMs is approximated through log-likelihood differences $\exp(\phi^{\pi_\theta} - \phi^{\pi_{\text{old}}})$. Notably, during the training of diffusion-based GRPO, the generation of completions for rewards calculation follows the exact block-based paradigm described in Section 2.1 using a fixed pre-defined block size $c$.

## 3. Proposed Method: *b1*

This section details *b1* as in Figure 4, compared with conventional fixed-size baselines. Specifically, *b1* consists of three components: (i) Dynamic-size Blocks, (ii) Monotonic Entropy Descent objective, and (iii) Model inference.

### 3.1. Dynamic-size Reasoning Blocks

**Dynamic Alignment of Blocks with Reasoning Boundaries.** The core of *b1* is the dynamic-size reasoning block $\mathbf{b}_k^d$ that aligns with the reasoning step, where $d$ indicates an adaptive size (compared to fixed-size $c$ in Section 2) and $k$ denotes the $k$-th block index. To support the dynamic size, a special indicator token $\tau_{\text{end}}$ is introduced to indicate the end of each reasoning step. Specifically, let $\hat{\mathbf{x}}_t$ denote the denoised sequence state at a discrete diffusion step $t$. For the $k$-th block, let $S_{k-1}$ denote the cumulative length of all preceding blocks (with $S_0 = 0$). The dynamic size $d$ is determined by the first appearance of the predicted indicator token $\tau_{\text{end}}$ within the sequence length limit $L$:

$$d = \min\{j \mid 1 \leq j \leq L, \hat{\mathbf{x}}_t[S_{k-1} + j] = \tau_{\text{end}}\}, \tag{4}$$

where $j$ is defined as the local index within the current block, and $S_{k-1} + j$ maps it to the global index in the sequence. In other words, once $\tau_{\text{end}}$ is generated, all masked tokens preceding $\tau_{\text{end}}$ are encapsulated within the current block, while any subsequent tokens are allocated to the following block. This adaptive mechanism ensures that each dynamic-size block $\mathbf{b}_k^d$ focuses on parallel generating a single, complete reasoning step, where the size $d$ varies dynamically with each reasoning step, thereby preventing disruption of the reasoning trace.

**Block Ending Indicator Reward.** To ensure the accurate generation of ending indicators and sufficient reasoning steps, an indicator reward $R_{\text{ind}}$ is proposed. The intuition is that complex reasoning inherently requires breaking down a problem into multiple intermediate steps (Tang et al., 2026). Thus, $R_{\text{ind}}$ promotes the model to produce a sufficient number of dynamic-size reasoning blocks rather than trivial, short chains. Let $K$ denote the total number of blocks, each of which contains a reasoning step, and $K_{\text{target}}$ be a pre-defined target value to avoid extreme cases with an excessive number of indicators. Therefore, the block ending indicator reward scales logarithmically with the number of blocks to encourage multi-step reasoning, defined as:

$$R_{\text{ind}} = \begin{cases} 1.0 & \text{if } K \geq K_{\text{target}} \\ \frac{\log(K+1)}{\log(K_{\text{target}}+1)} & \text{otherwise} \end{cases} \quad (5)$$

Intuitively, the indicator reward facilitates the generation of a rich chain of thought composed of sufficient reasoning blocks of varying sizes.

### 3.2. Training with Block Monotonic Entropy Descent

**Block Entropy Formulation.** With these dynamic-size blocks, the primary objective of *b1* is to encourage a monotonic entropy descent among the blocks as motivated in Figure 3. Following the mean-field formulation in block-based dLLMs (Zhao et al., 2025; Tang et al., 2025), which assumes token independence within a parallel decoding step, we define the block entropy $\mathcal{H}(\mathbf{b}_k^d)$ as the mean token-wise Shannon entropy within a dynamic-size block. Specifically, let $t^*$ denote the diffusion step where the indicator is generated, the block entropy is defined as:

$$\mathcal{H}(\mathbf{b}_k^d) = \frac{1}{d} \sum_{j=1}^{d} \mathcal{H}\left(\pi_\theta(\cdot \mid \hat{\mathbf{x}}_{t^*})_{S_{k-1}+j}\right), \quad (6)$$

where $\hat{\mathbf{x}}_{t^*}$ represents the denoised sequence at step $t^*$, $\mathcal{H}(\cdot)$ denotes the Shannon entropy over the full vocabulary distribution, and the subscript $S_{k-1} + j$ retrieves the probability distribution for the corresponding token in the sequence.

Although this formulation averages marginal token-wise entropies and does not explicitly model intra-block dependencies, we empirically observe that it functions as a reliable proxy for reasoning quality (Figure 3): lower block entropy

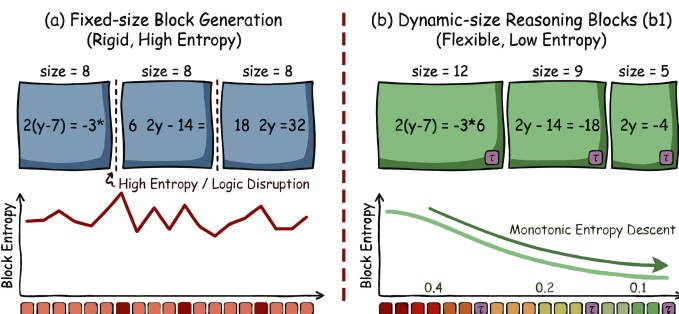

*Figure 4.* **Illustration of (a) conventional fixed-size block generation, and (b) our proposed dynamic-size reasoning blocks, for dLLMs.** Fixed-size generation enforces rigid sequence partitioning, which results in the generation of nondeterministic tokens with high entropy and yields incorrect results. In contrast, dynamic-size reasoning blocks align blocks with each reasoning step flexibly, avoiding the disruption of logical flow and leading to a monotonic descent trend of block-level entropy with correct results.

consistently aligns with more stable and logically coherent transitions, leading to accurate outcomes. Consequently, this block entropy serves as an indicative signal for learning reasoning-coherent blocks.

**Monotonic Entropy Descent with RL.** To facilitate monotonic entropy descent (MED) across the generated dynamic-size reasoning blocks via RL, an ideal objective is to minimise Spearman's Rank Correlation Coefficient, which serves as a global metric to quantify monotonicity. Based on this, we propose the negative Spearman's coefficient for block entropy, defined as:

> **Definition 1: Negative Spearman's Rank Correlation Coefficient for Block Entropy**
>
> Let $\{\mathcal{H}(\mathbf{b}_1^d), \ldots, \mathcal{H}(\mathbf{b}_K^d)\}$ denote a sequence of block entropies. Based on such a sequence, the negative Spearman's Rank Correlation Coefficient $r_{\text{SCC}}$ is defined as:
>
> $$r_{\text{SCC}} = -\left(1 - \frac{6\sum_{k=1}^{K} \delta_k^2}{K(K^2-1)}\right), \quad (7)$$
>
> where $\delta_k = \text{rank}(k) - \text{rank}(\mathcal{H}(\mathbf{b}_k^d))$ represents the difference between the rank of the block index $k$ with size $d$ and the rank of its corresponding entropy score.

Intuitively, $r_{\text{SCC}}$ quantifies the monotonicity of entropy changes throughout the generated reasoning blocks, where $r_{\text{SCC}} > 0$ indicates a monotonic descent trend in entropy. Yet, direct optimisation of this metric is intractable. Since Spearman's coefficient relies on the global rank operation, the reward signal for any specific block becomes coupled with the entropy values of all other blocks. Such a global dependency creates a highly volatile reward landscape that would induce high variance and unstable update directions in policy gradients, causing the optimisation to stagnate.

**Relaxation.** To overcome these optimisation challenges, a surrogate block entropy reward $R_{\text{ent}} \in [0, 1]$ is proposed. While the global Spearman's coefficient serves as the ultimate goal, enforcing the local entropy drop between adjacent blocks serves as a sufficient condition for MED that yields significantly denser feedback for RL training. By shifting the optimisation focus from a sparse global objective to these local pairwise comparisons, $R_{\text{ent}}$ effectively promotes a consistent entropy descent:

$$R_{\text{ent}} = \frac{1}{K-1} \sum_{k=2}^{K} \mathbb{I}\left(\mathcal{H}(\mathbf{b}_{k-1}^d) > \mathcal{H}(\mathbf{b}_k^d)\right), \quad (8)$$

where $\mathbb{I}(\cdot)$ denotes a binary indicator function. By decomposing the metric into independent pairwise comparisons, this surrogate provides fine-grained block-wise feedback for the training process and avoids the instability of global ranking optimisation. Crucially, we theoretically establish that our surrogate objective shares the same global optimal solution as Spearman's Rank Correlation Coefficient:

---

**Theorem 1: Equivalence between Block Entropy Reward and Spearman's Rank Correlation Coefficient**

The maximisation of the local reward $R_{\text{ent}}$ (Eq. (8)) between adjacent block pairs is mathematically equivalent to maximising the global negative Spearman's Rank Correlation Coefficient ($r_{\text{SCC}}$):

$$\arg\max R_{\text{ent}} = \arg\max r_{\text{SCC}}. \quad (9)$$

As the proposed $r_{\text{SCC}}$ measures the monotonic descent of block entropy, optimising the reward effectively promotes a strict block-based monotonic entropy descent.

---

Detailed proof is deferred to Appendix D. Intuitively, Theorem 1 suggests that maximising the surrogate entropy reward $R_{\text{ent}}$ effectively encourages the generation of dynamic-size reasoning blocks with strictly monotonic descending entropy. This indicates a more confident reasoning generation based on previous blocks, facilitating a coherent block-wise reasoning process. This conclusion can be further verified by the case studies in Table 9 and 10 in Appendix E.2.

The overall reward for RL training is defined as the aggregation of the proposed entropy and indicator rewards with other standard task-specific metrics (e.g., correctness and format rewards specific to each reasoning task following *d1* (Zhao et al., 2025) and *wd1* (Tang et al., 2025):

$$R_{\text{total}} = \alpha R_{\text{ent}} + \beta R_{\text{ind}} + \gamma R_{\text{task}}, \quad (10)$$

where $\alpha$, $\beta$, and $\gamma$ denote the reward weighting coefficients. Consequently, the final GRPO Training for *b1* integrates this total objective into the diffusion-based GRPO framework defined in Eq. (3). Following prior RL frameworks (Zhao et al., 2025; Tang et al., 2025), we set all weights to 1 (i.e.,

$\alpha = \beta = \gamma = 1$), which eliminates the need for hyper-parameter tuning and allows for seamless integration into existing RL frameworks. Further experiments on hyper-parameter sensitivity are provided in Appendix E.1. The overall training procedure is illustrated in Algorithm 1.

### 3.3. Inference with Dynamic-size Reasoning Blocks

The inference phase strictly aligns with the training phase in Section 3.1. Unlike the fixed-size block inference, *b1* dynamically generates the boundary of the current block $\mathbf{b}_k^d$ during the denoising process via the generated indicator $\tau_{\text{end}}$, which determines the end of the current block. Subsequently, the starting point of the next block is dynamically updated to the location immediately following the indicator to align with the next reasoning step. This adaptive generation procedure inherently anchors each block-wise parallel generation to a complete semantic reasoning step while avoiding any interruption, thereby preserving the logical integrity of each reasoning step and facilitating a more coherent and logically consistent reasoning trace output. The detailed inference procedure is illustrated in Algorithm 2.

### 3.4. Complexity Analysis

The computational overhead of *b1* is evaluated upon the fixed-size block baseline. For a generated sequence of $K$ blocks, the dynamic boundary construction for a sequence of size $L$ across $T$ diffusion steps yields a complexity of $\mathcal{O}(K \cdot T \cdot L)$. Combined with the block entropy calculation which scales linearly with the sequence length $L$, the total complexity is $\mathcal{O}(K \cdot T \cdot L + L)$. Such overhead is negligible compared to the complete dLLM generation process, which is dominated by self-attention that scales quadratically with the sequence length. Empirical results in Table 4 further verify that *b1* does not introduce significant overhead.

## 4. Experiments

**Setup.** To rigorously evaluate reasoning capabilities in a consistent setting, **our dataset and model selection are strictly aligned with *d1* (Zhao et al., 2025) and *wd1* (Tang et al., 2025).** Specifically, RL post-training is conducted on LLaDA-8B-Instruct (Nie et al., 2025) across four representative reasoning benchmarks: GSM8k (Cobbe et al., 2021), MATH (Lightman et al., 2023), Sudoku (Arel, 2025), and Countdown (Pan et al., 2025). Dataset splits are consistent with *d1* and *wd1*. Zero-shot pass@1 results are reported for sequence lengths of 256 and 512 tokens. Unless otherwise specified, *b1* will add $R_{\text{total}}$ to the default RL objective of *wd1*, within 256-token generations. Following existing protocols (Zhao et al., 2025; Tang et al., 2025), the best-performing checkpoints are reported for all experiments.

**Baseline.** Our baselines consist of the following categories. For RL frameworks for dLLMs, we integrate *b1* into state-of-the-art algorithms, including *Diffu*-GRPO (Zhao et al., 2025), GDPO (Rojas et al., 2025) and *wd1* (Tang et al.,

*Table 1.* **Overall performance on reasoning benchmarks.** Test accuracy is highlighted by **best** and runner-up. Δ shows the improvement by *b1* over base RL algorithms. The maximum generation length is configured as either 256 or 512.

| Model | Sudoku | | Countdown | | GSM8K | | MATH500 | |
|---|---|---|---|---|---|---|---|---|
| | 256 | 512 | 256 | 512 | 256 | 512 | 256 | 512 |
| LLaDA-8B-Instruct | 7.67 | 8.15 | 16.80 | 16.02 | 76.19 | 77.63 | 32.00 | 34.60 |
| + AdaBlock-dLLM | 6.69 | 7.93 | 14.84 | 16.80 | 76.27 | 77.94 | 32.40 | 33.60 |
| + SFT | 8.25 | 6.96 | 15.23 | 20.70 | 75.59 | 78.54 | 32.20 | 34.40 |
| + *Diffu*-GRPO | 13.53 | 11.94 | 19.92 | 24.22 | 76.35 | 79.98 | 33.60 | 36.80 |
| + *Diffu*-GRPO + *b1* | 16.97 | 18.85 | 28.91 | 32.03 | 78.39 | 80.97 | 34.60 | 37.20 |
| Δ | **+3.44** | **+6.91** | **+8.99** | **+7.81** | **+2.04** | **+0.99** | **+1.00** | **+0.40** |
| + GDPO | 9.94 | 10.38 | 17.97 | 20.70 | 76.57 | 79.23 | 32.20 | 35.00 |
| + GDPO + *b1* | 12.35 | 12.96 | 24.22 | 25.78 | 77.71 | 80.06 | 33.80 | 35.40 |
| Δ | **+2.41** | **+2.58** | **+6.25** | **+5.08** | **+1.14** | **+0.83** | **+1.60** | **+0.40** |
| + *d1* | 15.06 | 18.24 | 25.39 | 26.17 | 77.03 | 80.67 | 33.40 | 37.80 |
| + *d1* + *b1* | 18.48 | 23.12 | 30.47 | 34.38 | 78.24 | **81.73** | 34.40 | **38.60** |
| Δ | **+3.42** | **+4.88** | **+5.08** | **+8.21** | **+1.21** | **+1.06** | **+1.00** | **+0.80** |
| + *wd1* | 23.14 | 22.92 | 39.45 | 38.67 | 78.85 | 81.05 | 34.20 | 37.40 |
| + *wd1* + *b1* | **27.29** | **25.00** | **58.98** | **55.47** | **80.82** | 81.65 | **37.40** | 38.40 |
| Δ | **+4.15** | **+2.08** | **+19.53** | **+16.80** | **+1.97** | **+0.60** | **+3.20** | **+1.00** |

2025), all of which are currently limited to fixed-size block generation. Regarding the inference-time method, we compare against AdaBlock-dLLM (Lu et al., 2025), which is a recent tuning-free approach that truncates block generations at newline characters based on pre-defined confidence thresholds. Finally, for SFT, we follow the procedure in LLaDA (Nie et al., 2025) where the model undergoes supervised fine-tuning using the standard denoising cross-entropy objective without the involvement of RL.

**Implementation.** For a fair comparison with baselines, all methods are reproduced on four AMD MI300X GPUs (192GB) with a per-device batch size of 12 and gradient accumulation steps of 1 to achieve a consistent global batch size with the existing framework (Zhao et al., 2025; Tang et al., 2025; Rojas et al., 2025). The default block-ending indicator token is \block, and the default target block number $K_{target}$ is 10 for all experiments. For *wd1* (Tang et al., 2025), the KL term in GRPO is omitted to eliminate the dependency on $\pi_{ref}$ following their papers. Further implementation details are provided in Appendix C.

### 4.1. Overall Performance

The overall performance of *b1* is evaluated in Table 1. *b1* **demonstrates remarkable reasoning improvement for dLLMs** with a significant performance gain on the Countdown dataset of up to 19.53% with the *wd1* algorithm. This showcases the effectiveness of our proposed dynamic-size reasoning blocks, learnt by MED, in addressing the fixed-size block limitations in dLLMs for reasoning tasks. It is also clear that with dynamic-size reasoning block generation, all existing methods show consistently better performance compared to their default fixed-size block generation. This showcases that our proposed reward for block learning can effectively improve existing RL methods for dLLMs. Notably, SFT follows LLaDA (Nie et al., 2025) and operates at the token level without block-based generation during

the training process. Therefore, it does not significantly improve the base model performance, as there is a gap between training and block-based inference.

Besides RL training methods, we further compare with AdaBlock-dLLM, which is a recent inference-time method that truncates each block generation at high-confidence newline characters without learning dynamic-size blocks. It was originally designed for 4- or 5-shot scenarios. When reproduced in a consistent 0-shot inference setting alongside all other methods, it does not improve reasoning performance, as the reasoning step does not always align with the newline character and the model lacks the ability to learn dynamic-size blocks. We provide ablation studies and efficiency analysis in Section 4.2 and 4.7. Meanwhile, we provide in-depth explainable analysis of how MED correlates with reasoning performance in Section 4.3 and why *b1* improves reasoning by enhancing MED in Section 4.4 and 4.5. More analysis on **hyperparameter sensitivity, case studies on how *b1* improve reasoning coherence and reward dynamics are provided in Appendix E.**

*Table 2.* **Ablation study.** w/o denotes without $R_{ent}$.

| RL | Method | Sudoku | Countdown | GSM8K | MATH500 |
|---|---|---|---|---|---|
| *d1* | Fixed-size | 15.06 | 25.39 | 77.03 | 33.40 |
| | *b1* (w/o MED) | 15.65 | 26.17 | 78.01 | 34.20 |
| | *b1* (w/o $R_{ind}$) | 17.36 | 28.91 | 78.17 | 34.20 |
| | ***b1* (Full)** | **18.48** | **30.47** | **78.24** | **34.40** |
| *wd1* | Fixed-size | 23.14 | 39.45 | 78.85 | 34.20 |
| | *b1* (w/o MED) | 24.12 | 44.14 | 79.23 | 35.00 |
| | *b1* (w/o $R_{ind}$) | 26.07 | 55.86 | 80.06 | 36.60 |
| | ***b1* (Full)** | **26.71** | **58.98** | **80.82** | **37.40** |

### 4.2. Ablation Study

The ablation study of *b1* is presented in Table 2. Two variants are investigated: (1) **w/o MED**, removing the block monotonic entropy descent reward ($R_{ent}$); (2) **w/o $R_{ind}$**, removing the ending indicator reward ($R_{ind}$). The results demonstrate that both $R_{ind}$ and $R_{ent}$ contribute to *b1*. No-

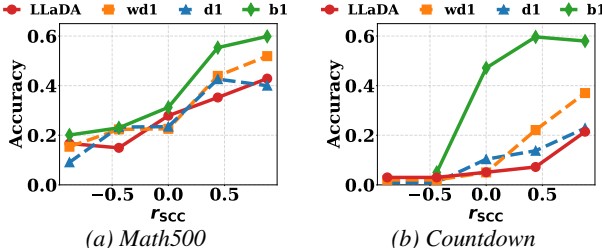

*(a) Math500*  *(b) Countdown*

*Figure 5.* **Analysis of MED with dLLMs' reasoning performance.** All reasoning tasks are ranked and grouped according to their $r_{SCC}$. Results demonstrate that reasoning generations with a higher $r_{SCC}$ (a monotonic descending trend in block entropy) achieve higher accuracy.

tably, removing the MED objective via $R_{ent}$ significantly degrades performance on the Countdown benchmark, indicating that the effectiveness of *b1* largely stems from facilitating monotonic entropy descent across the generated blocks. Meanwhile, removing $R_{ind}$ also leads to a performance decline, demonstrating that learning the block ending indicator to dynamically align block boundaries with semantic reasoning steps is critical for effective reasoning in dLLMs. **Overall, all variants of *b1* outperform their corresponding fixed-size block baselines, showcasing the effectiveness of our dynamic-size reasoning blocks.**

### 4.3. Correlation between MED and Reasoning

To investigate the relationship between Monotonic Entropy Descent (MED) and reasoning performance, we group all generated reasoning samples based on their $r_{SCC}$ values and evaluated the average accuracy for each bin. As illustrated in Figure 5, there is a strong positive correlation: **reasoning traces with a higher and positive $r_{SCC}$, which indicates a monotonic block entropy descending trend, yield higher reasoning performance.** This verifies the correlation between MED and reasoning. Notably, for Countdown, while fixed-size baselines produce a cluster of samples with negative $r_{SCC}$ (below -0.5), indicating an ascending entropy trend and resulting in near-zero accuracy. *b1* eliminates such low-$r_{SCC}$ groups and shifts the distribution towards higher $r_{SCC}$, verifying its ability to enhance monotonic entropy descent.

### 4.4. Effectiveness of *b1* in Improving MED

This experiment evaluates the effectiveness of *b1* in improving reasoning by enhancing Monotonic Entropy Descent (MED). As presented in Table 3, the experiments are conducted by measuring the average $r_{SCC}$ across all prompt questions and the proportion of block generations exhibiting a positive $r_{SCC}$ by different methods. Note that the Sudoku benchmark is excluded, as existing evaluation protocols (Zhao et al., 2025; Tang et al., 2025; Rojas et al., 2025) measure the overall proportion of correctly predicted cells rather than the correctness of individual questions. The results demonstrate that *b1* consistently achieves higher $r_{SCC}$ and $r_{MED}$ scores with superior reasoning performance compared to fixed-size baselines. **This indicates that our**

*Table 3.* **Effectiveness of *b1* in improving MED.** $r_{SCC}$ denotes the average Block-based Negative Spearman's Rank Correlation Coefficient (%), and $r_{MED}$ represents the proportion of block generations with a positive $r_{SCC}$ (%), reflecting a monotonic descending trend in block entropy.

|  | Countdown | | | GSM8K | | | MATH500 | | |
|---|---|---|---|---|---|---|---|---|---|
|  | $r_{SCC}$ | $r_{MED}$ | Acc. | $r_{SCC}$ | $r_{MED}$ | Acc. | $r_{SCC}$ | $r_{MED}$ | Acc. |
| *d1* | 57.30 | 90.23 | 25.39 | 47.99 | 87.87 | 77.03 | 18.67 | 67.80 | 33.40 |
| *wd1* | 58.98 | 91.41 | 39.45 | 33.13 | 79.00 | 78.85 | 17.93 | 65.80 | 34.20 |
| *b1* | 62.77 | 97.66 | 58.98 | 60.29 | 93.48 | 80.82 | 28.27 | 75.20 | 37.40 |

**dynamic-size blocks effectively enforce a more strictly descending entropy trend, which directly correlates with the improvements in reasoning discussed in Section 4.3.**

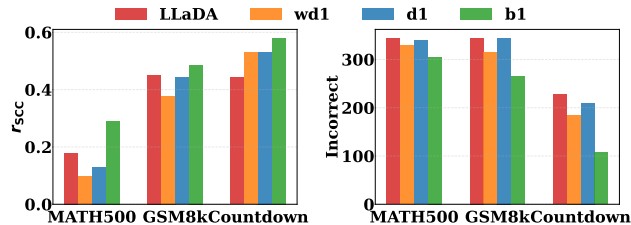

*Figure 6.* **Correlation between $r_{SCC}$ improvement and error reduction on hard reasoning samples.** For incorrectly predicted samples in the fixed-size baseline, *b1* demonstrates a unique capability to consistently increase $r_{SCC}$, thereby facilitating a monotonic entropy descent trend and reducing the reasoning errors.

### 4.5. *b1* in Improving Hard Reasoning Samples

This experiment delves into the "hard" reasoning samples, which are predicted incorrectly by the base LLaDA with fixed-size block generation. We analyse the $r_{SCC}$ and correctness for these hard samples by different methods in Figure 6. The results demonstrate that *b1* consistently enhances the $r_{SCC}$ for those instances that the fixed-size baseline failed to predict. Consequently, ***b1* consistently reduces the number of incorrect samples within the hard reasoning group.** This provides an in-depth explanation for *b1*'s superior reasoning performance.

### 4.6. Case Study

In this section, we further demonstrate the impact of fixed-size blocks on reasoning coherence in comparison with our proposed dynamic-size reasoning blocks by plotting the token entropy distribution as in Figure 7. Conventional fixed-block generation disrupts the mathematical calculation process and the logical flow because it results in repeated generation and non-coherent reasoning between adjacent blocks. These blocks often contain a vast amount of trivial tokens, such as full stops, with abnormally high entropy, which ultimately leads to an incorrect reasoning result. In contrast, our dynamic-size reasoning block incorporates each complete reasoning step within every single block to avoid rigid disruption and fragmentation of logic and calculation, thereby enhancing the coherence between each reasoning step. Overall, this study demonstrates a relatively lower entropy pattern characterised by descending entropy

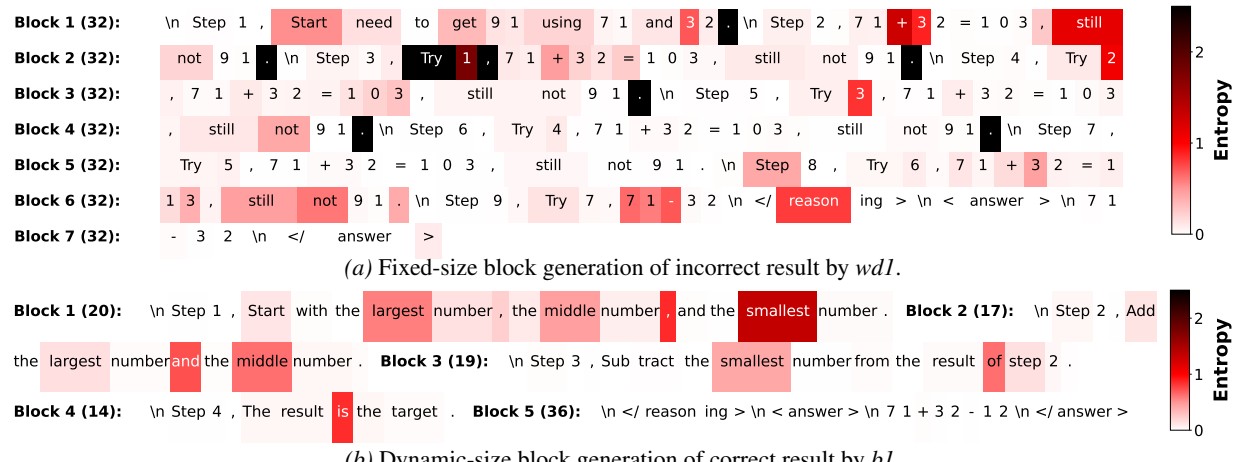

*(a)* Fixed-size block generation of incorrect result by *wd1*.

*(b)* Dynamic-size block generation of correct result by *b1*.

*Figure 7.* **Generation of *wd1* and our *b1*.** While fixed-size blocks in *wd1* disrupt reasoning steps and repeatedly generate high-entropy trivial tokens (e.g., full stops), our strategy dynamically aligns boundaries with reasoning steps. This ensures a coherent reasoning trace, leading to correct results. Special tokens including \block and <|endoftext|> are hidden for improved readability.

trends leading to the correct reasoning result, which conventional fixed-size block baselines often fail to achieve. More cases are in Figure 9 in Appendix E.2 due to page limit.

*Table 4.* **Comparison of training and inference efficiency** on 4×AMD MI300X GPUs. The average training time per gradient update step (s/step) and inference throughput (tokens/s) with the average performance are reported.

| RL | Method | Training | Inference | Avg. Acc. |
|---|---|---|---|---|
| *d1* | Fixed-size | 2.76 | 28.57 | 37.72 |
| | b1 | 2.82 | 27.03 | 40.40 |
| *wd1* | Fixed-size | 1.31 | 28.57 | 43.91 |
| | b1 | 1.68 | 27.03 | 51.12 |

## 4.7. Efficiency Analysis.

This section evaluates the computational efficiency of *b1* compared to fixed-size baselines. We measure the training cost using the average time per gradient update step and assess the inference efficiency via generation throughput (tokens per second) on 4×AMD MI300X GPUs. As in Table 4, *b1* incurs negligible overhead during training and maintains a competitive inference throughput comparable to baselines, while delivering superior average performance.

*Table 5.* **Effectiveness on additional domains.** Code generation, world knowledge, and scientific comprehension tasks are evaluated. The fixed-size baseline uses block size 32.

| Method | HumanEval | MBPP | KodCode | MMLU | MMLU-Pro | ARC-E | ARC-C |
|---|---|---|---|---|---|---|---|
| LLaDA | 28.66 | 33.60 | 24.00 | 53.46 | 33.79 | 76.98 | 73.21 |
| *d1* | 29.88 | 34.40 | 25.20 | 55.56 | 34.68 | 77.23 | 75.60 |
| *d1 + b1* | 31.10 | 35.00 | 27.00 | 58.98 | 36.32 | 80.93 | 78.67 |
| Improvement | +1.22 | +0.60 | +1.80 | +3.42 | +1.64 | +3.70 | +3.07 |
| *wd1* | 30.49 | 34.00 | 25.60 | 57.77 | 35.84 | 81.14 | 78.50 |
| *wd1 + b1* | 32.32 | 35.80 | 29.00 | 62.22 | 38.21 | 90.53 | 81.48 |
| Improvement | +1.83 | +1.80 | +3.40 | +4.45 | +2.37 | +9.39 | +2.98 |

## 4.8. Effectiveness on Additional Domains.

This section further evaluates the effectiveness of *b1* on additional domains, including code generation tasks, world knowledge tasks, and scientific comprehension tasks.

Specifically, MBPP, HumanEval, and KodCode are used for code generation, MMLU and MMLU-Pro are used for world knowledge evaluation, and ARC-E and ARC-C are used for scientific comprehension evaluation. For the fixed-size baselines, the default block size is set to 32, and the generation length is set to 256. For code generation, the \block tag is removed within the Python tag after generation to avoid syntax errors during testing. Since MBPP and HumanEval do not provide training splits, KodCodeLight-RL-10K (Xu et al., 2025) is used as the default GRPO training dataset for all code generation tasks, following GDPO, and 500 samples are randomly split as the test set for KodCode.

As shown in Table 5, *b1* consistently improves over fixed-size baselines across all evaluated tasks. Under *d1*, *b1* improves HumanEval, MBPP, and KodCode by 1.22, 0.60, and 1.80 points, respectively. Similar trends are observed under *wd1*, where *b1* achieves larger improvements, especially on KodCode, MMLU, and ARC-E. These results verify that *b1* effectively generalises beyond the mathematical domains.

*Table 6.* **Comparison with alternative intrinsic rewards.** Our proposed MED-based block entropy reward is compared with average block entropy reward, format reward, and block size reward across mathematical reasoning and code generation tasks.

| Method | MATH | GSM8K | HumanEval | MBPP |
|---|---|---|---|---|
| *wd1* (Fixed-size) | 34.2 | 78.9 | 30.5 | 34.0 |
| *b1* (MED Reward) | 37.4 | 80.8 | 32.3 | 35.8 |
| *b1* (Avg. Block Entropy Reward) | 34.6 | 79.3 | 29.9 | 34.4 |
| *b1* (Format Reward) | 33.8 | 78.2 | 28.7 | 33.6 |
| *b1* (Block Size Reward) | 34.4 | 79.4 | 30.5 | 33.2 |

## 4.9. Comparison with Alternative Rewards.

To isolate the effect of the MED-based reward, we compare it with three alternative intrinsic rewards. The first one is the average block entropy reward, computed as the negative average block entropy without MED. This reward directly encourages lower block entropy but does not model the

monotonic entropy decrease across reasoning blocks. The second is the format reward, computed according to the number of valid blocks, which encourages the model to generate more valid block structures. The third is the block size reward, computed as the negative block length, which controls block verbosity and encourages shorter blocks.

As shown in Table 6, the MED-based block entropy reward consistently outperforms all alternative rewards across mathematical reasoning and code generation tasks. Compared with *wd1* using fixed-size decoding, *b1* with the MED reward improves MATH from 34.2 to 37.4, GSM8K from 78.9 to 80.8, HumanEval from 30.5 to 32.3, and MBPP from 34.0 to 35.8. In contrast, directly minimising average block entropy only brings marginal improvements and even underperforms the fixed-size baseline on HumanEval. The format reward and block size reward also fail to provide consistent gains. These results indicate that simply encouraging lower entropy, more blocks, or shorter blocks is insufficient. The main improvement comes from the MED-based block entropy reward, which better captures the desired progressive reduction of uncertainty across reasoning blocks.

*Table 7.* **Illustration of reward hacking caused by excessive block indicators.** A simple example shows how inappropriate control of block indicators can fragment the reasoning process.

| Setting | Generated Reasoning | Effect |
|---|---|---|
| *b1* with proper $K$ | Step 1: The value increases from 5000 to 5125. \block Step 2: The profit is 5125 - 5000 = 125. \block | Coherent |
| Extremely Large $K$ | Step 1: The value is \block 5000 \block and increases to \block 5125 \block | Over-segmented |
| Without controlling $K$ | Step \block 1 \block value \block 5000 \block 5125 \block | Reward hacking |

### 4.10. Reward Hacking Analysis.

This section analyses the necessity of controlling the number of block indicators in *b1*. As shown in Table 7, when $K$ is properly constrained, the generated block boundaries are aligned with coherent reasoning steps. Each block corresponds to a meaningful intermediate step, such as computing the increased value and then deriving the final profit.

However, when $K$ is set to an excessively large value, the model tends to insert unnecessary block indicators inside a single reasoning step. This leads to over-segmentation, where one coherent computation is split into several fragmented pieces. The issue becomes more severe when $K$ is not controlled. In this case, the model can increase the intrinsic reward by repeatedly generating block indicators, even though the resulting reasoning chain becomes unnatural and less meaningful. These observations show that the improvement of *b1* does not come from simply generating more blocks. Instead, the block indicators need to be properly constrained so that they encourage coherent reasoning segmentation rather than reward hacking.

## 5. Related Work

This work intersects with three major research domains. **(i) Diffusion Large Language Models (dLLMs)** offer a potent alternative to autoregressive models by enabling parallel token generation by iterative denoising (Zhu et al., 2025a; Ye et al., 2025). Most dLLMs like LLaDA (Nie et al., 2025) typically adopt semi-autoregressive inference with fixed-size blocks (Arriola et al., 2025), where a group of tokens is generated in parallel at each step. Although this improves decoding efficiency, static block sizes are often insufficient for reasoning tasks with diverse semantic granularity, since different reasoning step may require distinct lengths (Xiong et al., 2025). **(ii) Reinforcement Learning (RL)** (Schulman et al., 2017) has been widely used to enhance the reasoning ability of LLMs. In particular, Group Relative Policy Optimisation (GRPO) (Shao et al., 2024) simplifies advantage estimation by using group-level rewards without a separate value model (Ouyang et al., 2022; Mroueh, 2025; Zhu et al., 2025b; Chen et al., 2025; Gutmann & Hyvärinen, 2012). **(iii) RL for dLLMs** is an evolving frontier. Existing frameworks such as MMaDA (Yang et al., 2025), *d1* (Zhao et al., 2025), LLaDA-DPO (Nie et al., 2025), and *wd1* (Tang et al., 2025) optimise dLLMs through token-level probability estimation or policy-ratio approximation, while methods such as SEPO (Zekri & Boullé, 2025), GDPO (Rojas et al., 2025), and TraceRL (Wang et al., 2025b) further improve stability or reduce variance (Zhu et al., 2026). However, these methods remain strictly confined to fixed-size block generation. Recent inference-time approaches such as AdaBlock-dLLM (Lu et al., 2025) and Deferred Commitment Decoding (Shu et al., 2026) introduce adaptive decoding heuristics, yet they focus mostly on non-learnable inference strategies and do not train dLLMs to produce reasoning-coherent dynamic blocks. In contrast, *b1* explicitly learns dynamic-size block generation through MED-based RL, bridging the gap between dLLM post-training and inference while enabling semantic alignment between blocks and reasoning steps. More details are provided in Section B.

## 6. Conclusion

In this paper, we identify that the rigid fixed-size block generation inherent in current Diffusion Large Language Models (dLLMs) hinders reasoning coherence, often characterised by non-monotonic entropy trends. To address this limitation, *b1* is proposed as the first RL framework to learn **dynamic-size blocks** aligned with semantic reasoning steps. Through a novel **Monotonic Entropy Descent (MED)** objective, *b1* facilitates coherent reasoning by ensuring progressively decreasing uncertainty. Extensive experiments verify that *b1* functions as an **effective plug-and-play method which consistently enhances reasoning performance compared to fixed-size block baselines.**

## Impact Statement

This paper presents b1, a novel RL-driven post-training framework designed to improve diffusion large language models (dLLMs) on reasoning tasks by learning dynamic-size reasoning blocks. This research relies exclusively on publicly available datasets and models, with all sources properly acknowledged. No direct negative societal impacts requiring specific safeguards or emphasis appear evident in the current study.

## Acknowledgments

This research has been supported by Australian Research Council Discovery Projects (DP230101196 and DE250100919).

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

# A. Appendix Overview

This appendix provides supplementary materials, theoretical proofs, and comprehensive experimental details to support the main findings of *b1*. We organised as follows:

- **Appendix B** reviews related literature, categorized into:

  ○ Diffusion Large Language Models in Appendix B.1;
  ○ Reinforcement Learning for Reasoning in Appendix B.2;
  ○ Reinforcement Learning for dLLMs in Appendix B.3;
  ○ dLLMs' Block-based Generation in Appendix B.4.

- **Appendix C** provides detailed implementation settings, covering:

  ○ Dataset Splits in Appendix C.1;
  ○ Reward Functions in Appendix C.2;
  ○ Model Training in Appendix C.3;
  ○ Reproducibility in Appendix C.4.

- **Appendix D** presents the rigorous mathematical proof for Theorem 3.2, establishing the equivalence between the proposed local entropy reward and the global Spearman's rank correlation coefficient.

- **Appendix E** offers additional experimental analysis, specifically:

  ○ Hyperparameter Sensitivity Analysis in Appendix E.1;
  ○ Visualisation of Dynamic Reasoning Blocks and Case Studies in Appendix E.2;
  ○ Training Reward Dynamics in Appendix E.3.
  ○ Discussion of Reward Difference between Existing dLLM-based RL Methods and *b1* in Appendix E.4.

# B. Related Work

## B.1. Diffusion Large Language Models

Diffusion Large Language Models (dLLMs) offer a new way to generate text compared to standard autoregressive (AR) models. Instead of generating one token after another, dLLMs generate text in parallel through a denoising process (Zhu et al., 2025a; Ye et al., 2025). Models like LLaDA (Nie et al., 2025) usually break the output into fixed-size blocks and generate these blocks in parallel (Arriola et al., 2025). However, using a fixed block size is not ideal because reasoning tasks vary in difficulty. Some steps need more tokens, while others need fewer (Xiong et al., 2025). A fixed size prevents the model from adjusting its resources based on the actual content.

## B.2. Reinforcement Learning for Reasoning

Reinforcement Learning (RL) is widely used to improve the reasoning skills of LLMs. A popular method is Group Relative Policy Optimisation (GRPO) (Shao et al., 2024). Unlike PPO (Schulman et al., 2017), which needs a complex value model, GRPO is simpler because it uses the average score of a group of outputs as a baseline (Ouyang et al., 2022). Recent studies show that GRPO is closely related to weighted regression (Mroueh, 2025). In fact, many RL methods for reasoning can be seen as a form of Rejection Sampling Fine-Tuning (RAFT) (Xiong et al., 2025). These methods work by increasing the probability of correct answers while pushing down the probability of wrong ones (Zhu et al., 2025b; Chen et al., 2025; Gutmann & Hyvärinen, 2012). This approach is often more stable than traditional RL and works well for reasoning problems where there are many ways to reach the right answer.

## B.3. Reinforcement Learning for dLLMs

Most recent research on applying RL to dLLMs focuses on direct optimisation methods. Early frameworks such as MMaDA (Yang et al., 2025), *d1* (Zhao et al., 2025), and LLaDA-DPO (Nie et al., 2025) work by estimating the probability of each token to guide the model. While this helps improve reasoning performance, estimating these probabilities is not always accurate, which can make the training process unstable. To avoid these issues, other methods like SEPO (Zekri & Boullé, 2025) use score-based objectives instead of relying on probability estimates. However, a key limitation remains across all

these approaches. Current methods, including *wd1* (Tang et al., 2025), GDPO (Rojas et al., 2025), and TraceRL (Wang et al., 2025b), are strictly limited to generating blocks of a fixed size. They do not allow the model to adjust the block size dynamically during the reasoning process, which is the main problem our method, *b1*, aims to solve.

### B.4. dLLMs' Block-based Generation

A very recent inference-time method termed AdaBlock-dLLM (Lu et al., 2025) was developed under few-shot scenarios to search for the next newline character and truncate blocks at that position. Despite providing variable block sizes, AdaBlock-dLLM operates as a hard-coded method reliant on pre-defined confidence thresholds and is restricted to splitting at newline characters, which do not always represent the true boundaries of reasoning steps.

Moreover, another very recent work under review, termed Deferred Commitment Decoding (DCD) (Shu et al., 2026), also presents a tuning-free inference-time method based on a confidence-aware sliding window to defer the high-uncertainty token until sufficient context is available. However, they leave a critical gap between dLLM training and inference, as they only focus on improving the dLLM inference stage on the fixed pre-trained model, while the decoding process during dLLM training or post-training is different from their proposed decoding strategies.

In contrast, *b1* focuses on a fundamentally different task by enabling dLLMs to explicitly learn the ability to generate reasoning-coherent and dynamic-size blocks via the proposed MED with RL, thereby bridging the gap between dLLM training and inference. Furthermore, in zero-shot reasoning scenarios following *d1* and *wd1*, we validate that the learnable *b1* achieves significant improvement, whereas the non-learnable AdaBlock-dLLM falls short across most tasks.

## C. Implementation Details

In this section, we provide further implementation details. **The full code is provided along with comprehensive scripts to reproduce all results with a single command.** Our provided code cover all evaluated methods, including the integration of $b1$ component into the $d1$, $wd1$, and $gdpo$ baselines.

### C.1. Dataset Split

To ensure a fair comparison, **our dataset selection and split strictly adheres to established protocols from prior research (Zhao et al., 2025; Tang et al., 2025).** For GSM8K, models undergo training on the standard training set and are subsequently assessed using the test split. For the Countdown benchmark, the training phase utilises the three-number subset sourced from TinyZero (Pan et al., 2025), while evaluation is conducted on a collection of 256 synthetic three-number problems established in previous work. The Sudoku experiments employ the $4 \times 4$ dataset, involving a training regime of one million distinct puzzles followed by testing on 256 synthetic instances. For MATH500, the official training partition serves as the basis for model training. As *b1* aims to align dLLM generation with semantic reasoning steps to enhance reasoning coherence, other domains such as code generation are excluded, as they may not necessitate explicit natural language reasoning chains targeted by *b1*.

### C.2. Reward Functions

**Besides our rewards, the task rewards employed in this study follow those utilised in *d1* (Zhao et al., 2025) and *wd1* (Tang et al., 2025),** with the specific configurations detailed below. Notably, all reward functions, including the reward introduced in *b1*, share the same reward weight 1.

**GSM8K.** Adopting the Unsloth reward architecture, the objective function incorporates five cumulative elements:

- **XML Structure Reward:** A credit of +0.125 is granted for each accurately placed tag, complemented by minor penalties for superfluous text following the tags.
- **Integer Answer Reward:** An allocation of +0.5 is assigned if the output constitutes a legitimate integer.
- **Correctness Reward:** A score of +2.0 is triggered when the final result aligns precisely with the verified ground truth.

**Countdown.** The evaluation framework consists of three distinct scenarios:

- A full credit of +1.0 is awarded if the formulated expression achieves the target value using the provided numerals.

- A partial credit of +0.1 is given if the numerals are utilised correctly but the target remains unreached.
- In all other instances, no reward is granted.

**Sudoku.** For Sudoku, the scoring system calculates the proportion of empty cells that are accurately populated, prioritising the model's problem-solving capability over simple data replication.

**MATH500.** For Math500, two additive sub-rewards facilitate the learning process:

- **Format Reward:** This component is tiered based on the output structure:
    - i) +1.00 for an `<answer>` tag containing a \boxed command;
    - ii) +0.75 for an `<answer>` tag alone;
    - iii) +0.50 for only the \boxed command;
    - iv) +0.25 for cases meeting neither condition.
- **Correctness Reward:** A value of +2.0 is provided if the precise solution is contained within the \boxed{} environment.

### C.3. Model Training

Our training protocols strictly adhere to the procedure established in *d1* (Zhao et al., 2025) and *wd1* (Tang et al., 2025). Implementation of GRPO is based on the TRL library on LLaDA-8b-Instruct, **ensuring that our model selection remains identical to prior methods (Zhao et al., 2025; Tang et al., 2025).** Note that other non-block dLLM backbones, such as Dream (Ye et al., 2025), are excluded as they do not originally designed with block-based generation paradigm. For the GRPO optimisation, Low-Rank Adaptation (LoRA) is integrated with a specific configuration of rank $r = 128$ and a scaling factor $\alpha = 64$ following *d1* (Zhao et al., 2025) and *wd1* (Tang et al., 2025).

The training of GRPO across the GSM8K, MATH, Countdown, and Sudoku benchmarks is performed on a cluster of four AMD MI300X GPUs. The experimental setup involves a sequence length of 256 tokens, a per-GPU batch size of 12, and a gradient accumulation frequency of 1. Each method undergoes training for a total of 10,000 global steps and is evaluated every 100 steps. Notably, *b1* reaches its peak performance significantly earlier than the baselines, specifically at step 1,000 for Sudoku, 2,100 for Countdown, 1,400 for MATH500, and 2,000 for GSM8K. In comparison, the state-of-the-art baseline *wd1* requires 1,500, 3,000, 1,500, and 5,000 steps to achieve optimal results, while *d1* exhibits a slower convergence rate, reaching its highest accuracy at 2,500, 7,500, 6,500, and 2,000 steps for the corresponding tasks.

Our model is optimised by the AdamW optimiser (Loshchilov & Hutter, 2017), utilising parameters $\beta_1 = 0.9$, $\beta_2 = 0.99$, and a weight decay coefficient of 0.1 following *d1* (Zhao et al., 2025) and *wd1* (Tang et al., 2025). The learning rate is fixed at $3 \times 10^{-6}$ with gradient clipping enforced at a threshold of 0.2. To enhance computational throughput, Flash Attention 2 (Dao, 2024) and 4-bit quantisation are employed. During gradient update cycles, random masking is applied to each prompt token with a probability of $p_{\text{mask}} = 0.15$ to facilitate log-probability estimation. More details on the hyperparameters are provided in Table 8.

### C.4. Reproducibility

The source code has been provided to facilitate reproduction with detailed instructions. We further provide pseudo-code for the training and inference procedure of *b1* in Algorithm 1 and 2.

*Table 8.* **Implementation details for reproducing results on b1, wd1, *d1*, and GRPO.** All hyperparameters follow the recommended values from their respective original papers. All experiments are conducted on four AMD MI300X GPUs, each equipped with 192 GB of VRAM.

| Parameter | *wd1* | *d1 & Diffu-GRPO* | GDPO |
|---|---|---|---|
| **Model and Precision** | | | |
| use_peft | true | true | true |
| torch_dtype | bfloat16 | bfloat16 | bfloat16 |
| load_in_4bit | true | true | true |
| attn_implementation | flash_attention_2 | flash_attention_2 | flash_attention_2 |
| lora_r | 128 | 128 | 128 |
| lora_alpha | 64 | 64 | 64 |
| lora_dropout | 0.05 | 0.05 | 0.05 |
| peft_task_type | CAUSAL_LM | CAUSAL_LM | CAUSAL_LM |
| **Training Configuration** | | | |
| seed | 42 | 42 | 43 |
| bf16 | true | true | true |
| sync_ref_model | True | True | True |
| ref_model_sync_steps | 64 | 64 | 64 |
| adam_beta1 | 0.9 | 0.9 | 0.9 |
| adam_beta2 | 0.99 | 0.99 | 0.99 |
| weight_decay | 0.1 | 0.1 | 0.1 |
| max_grad_norm | 0.2 | 0.2 | 0.2 |
| warmup_ratio | 0.0001 | 0.0001 | 0.0001 |
| learning_rate | 3e-6 | 3e-6 | 3e-7 |
| lr_scheduler_type | constant_with_warmup | constant_with_warmup | constant_with_warmup |
| **RL** | | | |
| num_generations | 6 | 6 | 6 |
| max_completion_length | 256 | 256 | 256 |
| max_prompt_length | 200 | 200 | 200 |
| block_length | 32 | 32 | 32 |
| diffusion_steps | 128 | 128 | 128 |
| generation_batch_size | 6 | 6 | 6 |
| remasking | low_confidence | low_confidence | low_confidence |
| random_masking | True | True | True |
| p_mask_prompt | 0.15 | 0.15 | 0.15 |
| beta | 0.00 | 0.04 | 0.00 |
| epsilon | – | 0.5 | 0.5 |
| num_iterations | 12 | 12 | 12 |

---

**Algorithm 1 RL Training for *b1***

---

**input** Dataset $\mathcal{D}$, Policy $\pi_\theta$, Reference $\pi_{\text{ref}}$, Steps $T$, Indicator $\tau_{\text{end}}$, Target $K_{\text{target}}$.
**output** Optimised Policy $\pi_\theta$.
1: **while** not converged **do**
2:     Sample batch of prompts $q \sim \mathcal{D}$.
3:     Generate $G$ completions $o_{1:G}$ using $\pi_\theta(\cdot|q)$.
4:     **for** each completion $o_g$ in $o_{1:G}$ **do**
5:         % — Reward Construction for $o_g$ —
6:         Initialise entropy list $\mathcal{H}_{\text{seq}} \leftarrow []$, block count $K \leftarrow 0$, cursor $S \leftarrow 0$.
7:         **while** $S < |o_g|$ **do**
8:             $K \leftarrow K + 1$.
9:             % Reconstruct reasoning steps from generated trace
10:            Identify dynamic size $d$ by locating first $\tau_{\text{end}}$.
11:            If no $\tau_{\text{end}}$ found, set $d$ to max block length.
12:            % Compute Block Entropy (Eq. (6))
13:            Compute $\mathcal{H}(\mathbf{b}_k^d)$ using token probabilities.
14:            Append $\mathcal{H}(\mathbf{b}_k^d)$ to $\mathcal{H}_{\text{seq}}$.
15:            Update cursor $S \leftarrow S + d$.
16:        **end while**
17:        % Compute Rewards
18:        $N_{\text{drop}} \leftarrow \sum_{k=2}^{K} \mathbb{I}\left(\mathcal{H}(\mathbf{b}_{k-1}^d) > \mathcal{H}(\mathbf{b}_k^d)\right)$.
19:        $R_{\text{ent}} \leftarrow N_{\text{drop}}/(K-1)$ (Eq. (8)).
20:        Compute $R_{\text{ind}}$ using $K, K_{\text{target}}$ (Eq. (5)).
21:        Obtain task reward $R_{\text{task}}$ (e.g., correctness).
22:        $R_{\text{total}} \leftarrow R_{\text{ent}} + R_{\text{ind}} + R_{\text{task}}$ (Eq. (10)).
23:    **end for**
24:    % GRPO Update
25:    Compute advantages $\hat{A}_g$ based on $R_{\text{total}}$ group statistics.
26:    Update $\pi_\theta$ via Eq. (3).
27: **end while**
28: **return** $\pi_\theta$

---

---

**Algorithm 2 Inference with Dynamic-size Reasoning Blocks**

---

**input** Prompt $\mathbf{x}_{\text{prompt}}$, Model $\pi_\theta$, Diffusion Steps $T$, Indicator $\tau_{\text{end}}$, Max Sequence Length $L$.
**output** Generated Reasoning Trace $\mathbf{y}$.
 1: Initialise sequence $\hat{\mathbf{x}}$ with $\mathbf{x}_{\text{prompt}}$ and masked tokens up to length $L$.
 2: Initialise cumulative generated length $S \leftarrow 0$.
 3: **while** $S < L$ **and** End-of-Sequence (EOS) not generated **do**
 4:   % Default dynamic size is the remaining length
 5:   Initialise dynamic size $d \leftarrow L - S$.
 6:   % Parallel Generation with Dynamic Boundary
 7:   **for** $t = T$ **to** $1$ **do**
 8:     Denoise tokens from index $S + 1$ to $L$ to obtain $\hat{\mathbf{x}}_t$ using $\pi_\theta(\hat{\mathbf{x}}_{t+1}, t)$.
 9:     % Check for indicator token to determine early exit
10:     Search for $\tau_{\text{end}}$ in $\hat{\mathbf{x}}_t$ within range $[S + 1, L]$.
11:     **if** $\tau_{\text{end}}$ generated at index $j$ **then**
12:       $d \leftarrow j - S$ (Set dynamic size relative to $S$).
13:       **break**
14:     **end if**
15:   **end for**
16:   % Commit the dynamic generation
17:   Finalise tokens $\mathbf{b}^d = \hat{\mathbf{x}}_0[S + 1 : S + d]$.
18:   Update sequence $\hat{\mathbf{x}}$ with finalised $\mathbf{b}^d$.
19:   % Move to next reasoning step
20:   $S \leftarrow S + d$.
21:   **if** EOS $\in \mathbf{b}^d$ **then**
22:     **break**
23:   **end if**
24: **end while**
25: **return** $\hat{\mathbf{x}}[1 : S]$

---

## D. Proof for Theoretical Insights

In this section, we provide a rigorous proof for Theorem 1.

---

**Theorem 1: Equivalence between Block Entropy Reward and Spearman's Rank Correlation Coefficient**

The maximisation of the local reward $R_{\text{ent}}$ (Eq. (8)) between adjacent block pairs is mathematically equivalent to maximising the global negative Spearman's Rank Correlation Coefficient ($r_{\text{SCC}}$):

$$\arg\max R_{\text{ent}} = \arg\max r_{\text{SCC}}. \tag{11}$$

As the proposed $r_{\text{SCC}}$ measures the monotonic descent of block entropy, optimising the reward effectively promotes a strict block-based monotonic entropy reduction.

---

*Proof.* Let $\mathbf{k} = [1, 2, \dots, K]$ denote the strictly increasing vector of block indices, and $\mathbf{r} \in \mathbb{R}^K$ be the vector of entropy ranks, where $r_k = \text{rank}(\mathcal{H}(\mathbf{b}_k^d))$.

First, we analyse the optimisation direction of the global metric. Based on the definition in Eq. (9), $r_{\text{SCC}}$ is linearly proportional to the sum of squared rank differences, $\sum \delta_k^2$. Since the coefficient $\frac{6}{K(K^2-1)}$ is positive, maximising $r_{\text{SCC}}$ is equivalent to maximising the squared differences:

$$\arg\max r_{\text{SCC}} = \arg\max \sum_{k=1}^{K} \delta_k^2. \tag{12}$$

By expanding $\delta_k^2 = (k - r_k)^2$, we isolate the variable component:

$$\sum_{k=1}^{K} \delta_k^2 = \underbrace{\sum_{k=1}^{K} k^2 + \sum_{k=1}^{K} r_k^2}_{\text{Constant } C} - 2\mathbf{k} \cdot \mathbf{r}. \tag{13}$$

Since the sum of squares for indices and ranks is a permutation-invariant constant $C$, maximising the sum of squared differences is algebraically equivalent to minimising the dot product $\mathbf{k} \cdot \mathbf{r}$:

$$\arg\max r_{\text{SCC}} = \arg\min(\mathbf{k} \cdot \mathbf{r}). \tag{14}$$

According to the Rearrangement Inequality (Li et al., 2025), the dot product of two vectors is minimised if and only if they are oppositely sorted. Since the block index vector $\mathbf{k}$ is fixed as strictly increasing ($1 < \cdots < K$), the global minimum of $\mathbf{k} \cdot \mathbf{r}$ is achieved if and only if the rank vector $\mathbf{r}$ is strictly decreasing ($r_1 > \cdots > r_K$). This rank order corresponds to a strictly monotonic descent in block entropy values:

$$\arg\min(\mathbf{k} \cdot \mathbf{r}) \implies \{\mathcal{H}_{\text{seq}} \mid \mathcal{H}(\mathbf{b}_1^d) > \mathcal{H}(\mathbf{b}_2^d) > \cdots > \mathcal{H}(\mathbf{b}_K^d)\}. \tag{15}$$

Next, we look at the local surrogate. $R_{\text{ent}}$ is the mean of binary indicators $\mathbb{I}(\mathcal{H}(\mathbf{b}_{k-1}^d) > \mathcal{H}(\mathbf{b}_k^d))$. Maximising $R_{\text{ent}}$ to its theoretical upper bound ($R_{\text{ent}} = 1$) requires the indicator condition to hold for all adjacent blocks $k$. By transitivity, this pairwise constraint forces the entire sequence to be strictly decreasing:

$$\arg\max R_{\text{ent}} \implies \{\mathcal{H}_{\text{seq}} \mid \mathcal{H}(\mathbf{b}_1^d) > \mathcal{H}(\mathbf{b}_2^d) > \cdots > \mathcal{H}(\mathbf{b}_K^d)\}. \tag{16}$$

Consequently, both the maximisation of the local reward $R_{\text{ent}}$ and the maximisation of the global metric $r_{\text{SCC}}$ converge to the same necessary and sufficient condition: a strictly monotonically decreasing entropy sequence:

$$\arg\max R_{\text{ent}} = \arg\max r_{\text{SCC}}. \tag{17}$$

$\square$

# E. Additional Experimental Results for *b1*

This section offers further experimental analysis for the proposed *b1* framework, incorporating comprehensive case studies, hyperparameter sensitivity analysis and visualisations for reward dynamics.

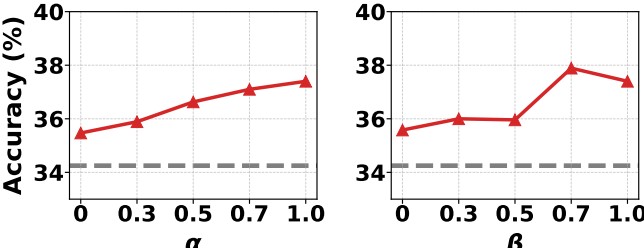

*Figure 8.* **Hyperparameter Sensitivity Analysis on MATH500.** The grey line represents the baseline *wd1* performance by fixed-size blocks, while the red line denotes *wd1 + b1* across varying reward weights. Results indicate that *b1* is robust to hyperparameter variations and consistently outperforms the fixed-size baseline. Furthermore, the performance gain at weight 1.0 compared to 0 confirms the contribution of the proposed rewards.

## E.1. Hyperparameter Sensitivity Analysis

The impact of hyperparameter variations on the reasoning performance of *b1* is evaluated on the MATH500 dataset. We vary the specific reward weight while keeping other hyperparameters fixed to assess their influence. The baseline *wd1* by the fixed-size block is denoted as the grey line, compared with *wd1 + b1*, which is represented by the red line. As shown in Figure 8, *b1* is not sensitive to reward weight changes and consistently outperforms the fixed-size block baseline *wd1*. Notably, the reasoning performance is superior at the default reward weight of 1.0 compared to 0, showcasing that our proposed indicator and entropy rewards effectively contribute to the overall effectiveness.

## E.2. Visualisation of Dynamic Reasoning Blocks Learnt by Monotonic Entropy Descent

This section provides a comprehensive visualisation for the proposed dynamic-size reasoning blocks on the Countdown mathematical reasoning task in Figure 7. The ground truth for this case study is 91 derived from 71 + 32 - 12. As illustrated in the qualitative results, the fixed-size baseline rigidly disrupts the logic flow and causes the model to lose track of previous steps. This fragmentation leads to repetitive generations and abnormally high entropy for trivial tokens such as full stops, which are highlighted in black. Such high entropy indicates non-deterministic generations where the dLLM struggles to conclude reasoning steps, ultimately yielding an incorrect answer like 71 - 32.

A similar case occurs for the GSM8k reasoning dataset in Figure 9. The fixed-size generation strategy rigidly partitions the reasoning logic flow and often truncates mathematical calculations, for instance, splitting the number 180 into 18 and 0 across adjacent blocks. This fragmentation triggers the generation of tokens with abnormally high entropy, such as full stops, 'so' and 'to', which indicates that the dLLM is non-deterministic and struggles to conclude reasoning steps.

In contrast, our proposed strategy dynamically aligns block boundaries with each reasoning step. **This alignment preserves logical coherence between blocks, avoids disrupting logic flows and calculation process and maintains a deterministic low-entropy pattern, thereby resulting in correct reasoning results.** Additional case studies in Table 9 and 10 further verify the effectiveness of dynamic-size blocks in maintaining reasoning coherence and achieving better reasoning results.

*Table 9.* **Illustration of How *b1* Improve Reasoning Coherence and Quality.** A case study from the MATH dataset compares the baseline *wd1* by default fix-size generation (size=32) and our dynamic-size reasoning blocks learnt by our MED. While both methods successfully formulate the initial slope equation, *wd1* with default fixed-size block generation, encounter a logical error during algebraic expansion. In contrast, the dynamic-size reasoning blocks in *b1* maintain strict logical consistency throughout the derivation to reach the correct solution. The raw generations are provided in Table 10

| Method | Fixed-size Block Generation | Dynamic-size Block Generation (Ours) |
|---|---|---|
| **Problem** | Given points $P(-2, 7)$ and $Q(4, y)$, for what value of $y$ is the slope equal to $\frac{-3}{2}$? | |
| **Step 1: Formulation** | Both models correctly identify the slope formula and set up the equation: $$\frac{y-7}{4-(-2)} = \frac{-3}{2} \implies \frac{y-7}{6} = \frac{-3}{2} \implies 2(y-7) = -18$$ | |
| **Step 2: Execution** | **Logical Error** The model fails to correctly distribute the multiplication: $2y - \mathbf{2} = -18$ (*Incorrect Expansion*) *Error:* $2 \times 7$ *was incorrectly calculated as* $2$. | **Coherent Reasoning** The model correctly applies the distributive property: $2y - \mathbf{14} = -18$ (*Correct Expansion*) *Result: The logical chain remains intact.* |
| **Step 3: Solution** | Solves the incorrect equation: $2y = -16 \implies y = -8$ | Solves the correct equation: $2y = -4 \implies y = -2$ |
| **Final Answer** | **-8** ✗ | **-2** ✓ |

*Table 10.* **Visualisation of the block ending indicator for our dynamic-size reasoning blocks on a MATH reasoning task.** The outputs are presented in their original format to illustrate the internal reasoning process. The block ending indicators $\tau_{\text{end}} = \backslash \texttt{block}$ accurately align with each reasoning step, and the dynamic-size reasoning blocks result in the correct solution.

| Fix-size Block Generation | Dynamic-size Block Generation |
|---|---|
| ```
<reasoning>
The slope of a line passing through
points (x1, y1) and (x2, y2) is given
by the formula:
slope = (y2 - y1) / (x2 - x1)
For points P(-2, 7) and Q(4, y),
the slope is:
slope = (y - 7) / (4 - (-2)) = (y - 7) / 6
We are given that this slope is equal
to -3 / 2. Therefore, we set up
the equation:
(y - 7) / 6 = -3 / 2
To solve for y, we cross-multiply:
2(y - 7) = -3 * 6
2y - 2 = -18
2y = -16
y = -8
</reasoning>
<answer>
-8
</answer>
``` | ```
<reasoning>
Step 1, The slope of a line through
two points (x1, y1) and (x2, y2)
is given by the formula (y2 - y1) / (x2 - x1). \block
Step 2, For points P(-2,7) and Q(4,y), the slope is
(y - 7) / (4 - (-2)) = (y - 7) / 6. \block
Step 3, We are given such to be equal
to -3/2, so we set up the equation
(y - 7) / 6 = -3/2. \block
Step 4, Solving for y, we cross-multiply
to get 2(y - 7) = -3 * 6, which
simplifies to 2y - 14 = -18. \block
Step 5, Adding 14 to both sides gives
2y = -4, and dividing by 2 gives
y = -2. \block
</reasoning>
<answer>
y = -2
</answer>
``` |

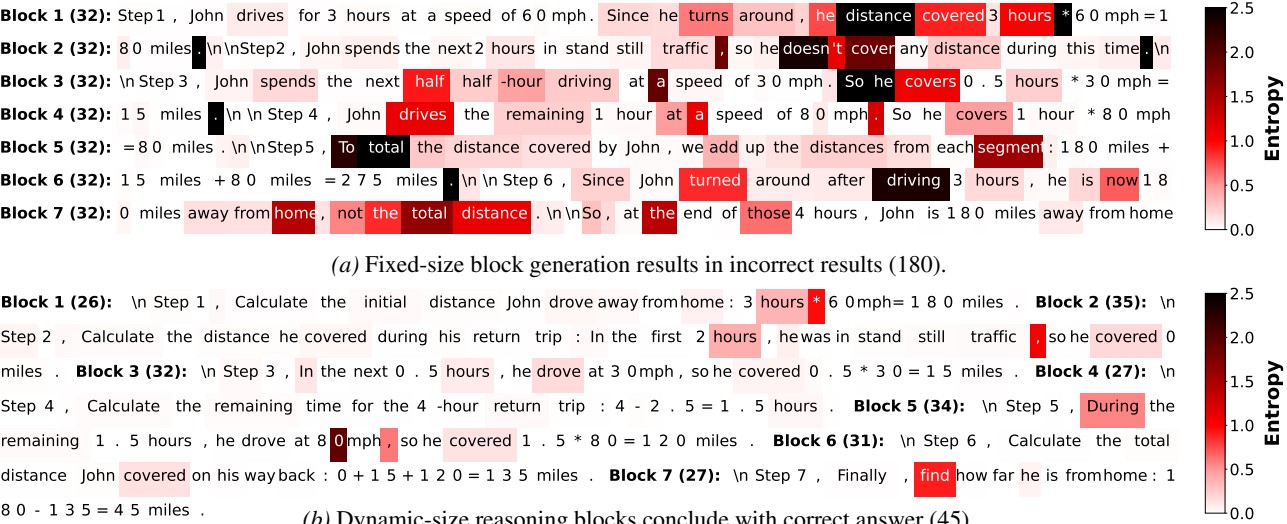

*(a)* Fixed-size block generation results in incorrect results (180).

*(b)* Dynamic-size reasoning blocks conclude with correct answer (45).

*Figure 9.* **Case studies on the GSM8K dataset compare dLLMs post-trained via *wd1* using default fixed-size blocks against proposed dynamic reasoning block generation strategies**, evaluated on the "John's driving" problem (Ground Truth: 45). The number behind each block indicates the block size. Fixed-size models rigidly partition reasoning logic and truncate mathematical calculations. Such fragmentation of the reasoning process triggers the generation of tokens with abnormally high entropy, even for trivial characters like full stops (tokens in black), indicating that the dLLM is non-deterministic and struggles to decide whether to conclude reasoning steps. In contrast, the proposed strategy dynamically aligns boundaries with reasoning steps to preserve logical coherence, avoiding high entropy tokens and ensuring correct final answers. Special tokens, including \block and <|endoftext|> remain hidden for readability.

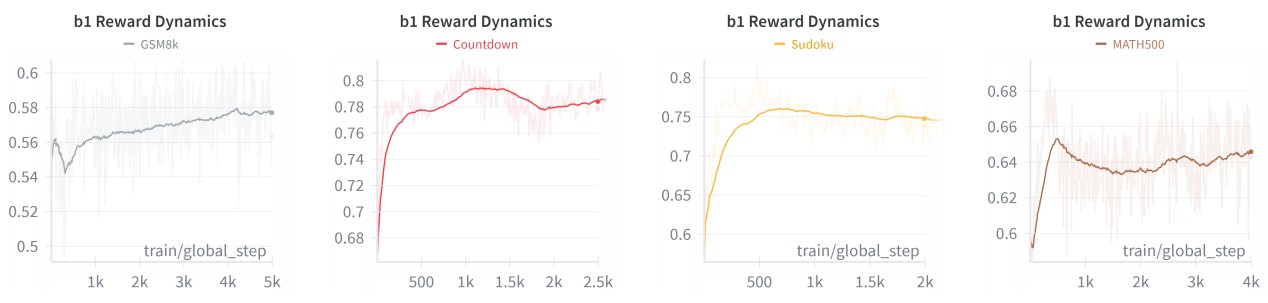

*(a)* GSM8k      *(b)* Countdown      *(c)* Sudoku      *(d)* Math500

*Figure 10.* Training Reward Dynamics in *b1*. The curves represent the sum of the proposed block entropy and block ending indicator rewards while being smoothed using a Time-Weighted Exponential Moving Average to highlight underlying trends. Results verify that the newly introduced rewards in *b1* converge mostly within the first 1000 gradient steps during training.

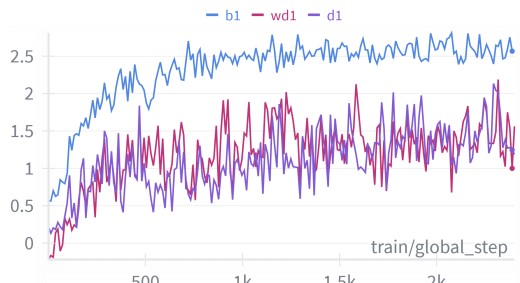

*Figure 11.* Comparison of Reward Dynamics between *b1*, *d1* and *wd1* on GSM8k. In contrast to Figure 10, the reward dynamics illustrated here represent the aggregation of all reward components. While *d1* and *wd1* exhibit fluctuating overall reward dynamics with their fixed-size block generation, *b1* demonstrates a more stable reward dynamics and faster convergence via its dynamic-size reasoning block learning. This highlights the efficacy of the dynamic-size block generation mechanism in *b1* for enhancing a more stable RL training for dLLMs.

### E.3. Training Reward Dynamics on *b1*

Figure 10 illustrates the training dynamics of the sum of the two proposed intrinsic rewards in *b1*: the block entropy reward $R_{\text{ent}}$ and the block ending indicator reward $R_{\text{ind}}$. It is evident that the aggregated reward curve exhibits rapid convergence, mostly stabilising within the first 1,000 gradient steps. This suggests that the model efficiently learns to generate the block ending indicator $\tau_{\text{end}}$ to delineate meaningful reasoning boundaries and quickly adapts to the objective of monotonic entropy descent. Such rapid convergence demonstrates the effectiveness of our rewards, which effectively guide the policy towards generating coherent dynamic-size blocks without requiring prolonged training periods.

Furthermore, Figure 11 compares the overall reward dynamics of *b1* against existing dLLM RL frameworks, *d1* and *wd1*, on the GSM8K dataset. The baselines, constrained by fixed-size block generation, exhibit significant volatility and fluctuations throughout the training process. This instability likely stems from the rigid block partitioning that disrupts logical flow, causing high variance in the global reward signals as discussed in Section 3. In contrast, *b1* demonstrates markedly superior stability and faster convergence. By learning dynamic-size reasoning blocks that align with the semantic flow, *b1* mitigates the optimisation difficulties associated with fixed rigid block generations, thereby establishing a more stable RL training paradigm for dLLMs.

### E.4. Discussion of Difference between Existing Reward Functions

Existing RL frameworks (Zhao et al., 2025; Tang et al., 2025; Rojas et al., 2025; Wang et al., 2025a; He et al., 2025) for dLLMs primarily utilise rewards at the sentence or answer level. For instance, their correctness and format rewards focus exclusively on whether the final answer matches the ground truth or adheres to pre-defined XML structures. While effective for outcome evaluation, such rewards provide sparse feedback and fail to account for the internal quality of the reasoning process.

In contrast, our proposed rewards in *b1* operate within the logit space at a fine-grained token and block level on the reasoning

generation. Specifically, the block entropy reward leverages the probability distributions of reasoning tokens to provide dense and continuous feedback during generation. By shifting the optimisation signal from external format matching to internal reasoning uncertainty reduction, *b1* enables the dLLMs to learn the coherent reasoning traces rather than merely memorising the correct final strings.

