# OpenReview forum: "Break the Block: Dynamic-size Reasoning Blocks for Diffusion Large Language Models via Monotonic Entropy Descent with Reinforcement Learning"
_ICML.cc/2026/Conference — ICML 2026 regular_

### Official Review · Reviewer_6XN7 · 2026-02-22

**Soundness:** 3
**Presentation:** 3
**Significance:** 3
**Originality:** 3
**Overall Recommendation:** 5
**Confidence:** 5

**Summary:**

This paper studies block-based inference in current diffusion language models (DLMs) and argues that fixed block sizes can harm reasoning by splitting semantic reasoning steps and breaking coherence. The authors propose b1, a post-training framework for DLMs that learns dynamic-size block boundaries by adding an end-of-block indicator and training its prediction with RL. The method combines an indicator reward to encourage producing a sufficient number of blocks, and introduces monotonic entropy descent to promote adjacent-block entropy decreases. Empirical results report consistent gains over multiple RL-trained baselines for DLMs.

**Compliance With Llm Reviewing Policy:**

Affirmed.

**Final Justification:**

I recommend accept for this paper. The paper tackles an important issue in block diffusion language models, ie the mismatch between fixed block-based decoding and the semantic structure of multi-step reasoning. The proposed solution is simple, well motivated, and practically useful, and the empirical gains are strong and consistent. The paper is also generally clear and well presented.

My main concerns were well addressed in the rebuttal. I also appreciate the additional clarification and case study in the follow-up response, which clearly illustrate the reward-hacking and over-segmentation behaviors and make the role of controlling K_\text{target} much clearer. This resolves my remaining concern on this point. Overall, the rebuttal strengthened the paper and increased my confidence in the method.

**Key Questions For Authors:**

- Have you evaluated b1 on code generation tasks (where the structural patterns may make dynamic blocks particularly beneficial)? If not, do you expect similar gains?
- Do you observe reward hacking behaviors with the indicator token $\tau_{end}$ (e.g., excessive or unnatural block segmentation)?
- What are the learned distribution / stats of dynamic block sizes across prompts or tasks?

**Limitations:**

Not quite; see weaknesses and questions above.

**Strengths And Weaknesses:**

Strengths:
- The paper clearly motivates the problem with intuitive examples: rigid fixed-width block boundaries can break reasoning traces, and entropy dynamics correlate with correctness. The proposed method targets an important mismatch between block-based decoding and the semantics of multi-step reasoning.
- The learning algorithm of explicit boundary indicators is a simple, modular technique and can integrate easily with existing RL post-training pipelines, with negligible additional overheads.
- Despite the simplicity, the empirical improvements are strong and consistent across baselines, suggesting the approach is practically useful.

Weaknesses:
- A fixed $K_\text{target}$ is used across prompts and tasks; while it works well as a regularizer to encourage a sufficient number of blocks, it probably biases the learned block-size dynamics. A more thorough analysis of sensitivity and guidance for this hyperparameter will strengthen the paper.
- The block-level entropy is a good indicator, but the specified estimation and assumptions might correlate with many factors beyond reasoning coherence, such as formatting, verbosity, tokenization artifacts, hacked behaviors, etc. More diagnostics and reward ablations would clarify the mechanism further.

---

> ### Author Rebuttal · Authors · 2026-03-27
>
> **We sincerely appreciate the constructive questions you've provided, which are greatly helpful for us to improve our paper.** We have thoroughly addressed your questions as follows:
>
> >**Re W1: Analysis of $K_\text{target}$**
>
> Following your suggestion, we have further examined the **sensitivity of $K_\text{target}$**:
>
> | $K_{\text{target}}$ | MATH | GSM8K | HumanEval | MBPP |
> | :---: | :---: | :---: | :---: | :---: |
> | **0** | 34.2 | 78.9 | 30.5 | 34.0 |
> | **5** | 36.8 | 80.2 | 31.1 | 36.0 |
> | **10** | 37.4 | 80.8 | 32.3 | 35.8 |
> | **20** | 36.2 | 79.5 | 31.1 | 35.0 |
> | **50** | 34.6 | 78.3 | 28.7 | 34.2 |
>
> Results show that **b1 is robust** across a large range of $K_\text{target}$, roughly from 5-20. We have added this into our manuscript.
>
> >**Re W2: Block Entropy and Other Rewards**
>
> We agree with you that there should be experiments to clarify the block-level entropy mechanism compared with other factors. We provided **case study in Fig 7 & 9 to qualitatively evaluate how reasoning coherence is correlated to block-level entropy**. Tab 6 & 7 show the **reasoning examples comparison** between b1 (optimised with block level entropy) and baselines, where the **reasoning coherence difference is significant**.
>
> Following your suggestions, we also conducted experiments by **replacing its MED-based block entropy reward with three alternative intrinsic rewards**: (i) **avg block entropy reward**, computed as the negative average block entropy (without MED), (ii) **format reward**, computed as the number of valid blocks (encouraging more blocks), and (iii) **block size reward**, computed as the negative block length (controlling block verbosity).
>
> |  | MATH | GSM8K | HumanEval | MBPP |
> | :--- | :---: | :---: | :---: | :---: |
> | **wd1 (fixed-size)** | 34.2 | 78.9 | 30.5 | 34.0 |
> | **b1 (MED reward)** | **37.4** | **80.8** | **32.3** | **35.8** |
> | **b1 (avg block entropy reward)** | 34.6 | 79.3 | 29.9 | 34.4 |
> | **b1 (format reward)** | 33.8 | 78.2 | 28.7 | 33.6 |
> | **b1 (block size reward)** | 34.4 | 79.4 | 30.5 | 33.2 |
>
> The results show that **MED-based block entropy reward gives the main improvement** compared to other intrinsic rewards, such as avg block entropy reward, which directly encourages low average block entropy. This further isolates MED as the key mechanism for b1's effectiveness. We have added this to our manuscript.
>
> >**Re Q1 Generalisation**
>
> Following your suggestions, we further evaluate b1 on **code generation** (MBPP, HumanEval and KodCode), **world knowledge** (MMLU and MMLU-Pro) and **scientific comprehension** (ARC-E and ARC-C) tasks.
>
> |Method|HumanEval|MBPP|KodCode|MMLU|MMLU-Pro|ARC-E|ARC-C|
> |:---|:---|:---|:---|:---|:---|:---|:---|
> |LLaDA|28.66|33.60|24.00|53.46|33.79|76.98|73.21|
> |d1|29.88|34.40|25.20|55.56|34.68|77.23|75.60|
> |**d1+b1**|**31.10**|**35.00**|**27.00**|**58.98**|**36.32**|**80.93**|**78.67**|
> |**${\Delta}$**|**+1.22**|**+0.60**|**+1.80**|**+3.42**|**+1.64**|**+3.70**|**+3.07**|
> |wd1|30.49|34.00|25.60|57.77|35.84|81.14|78.50|
> |**wd1+b1**|**32.32**|**35.80**|**29.00**|**62.22**|**38.21**|**90.53**|**81.48**|
> |**${\Delta}$**|**+1.83**|**+1.80**|**+3.40**|**+4.45**|**+2.37**|**+9.39**|**+2.98**|
>
> **b1 achieves consistent improvement on these new domains.**
>
> >**Re Q2 Potential Reward Hacking**
>
> Thank you for your insightful question. Our design of $R_{ind}$ in Eq. 5 ensures that **if too many blocks are generated, the reward will be capped at 1**. This is controlled by a threshold $K_{target}$. In this way, b1 prevents the potential indicator reward hacking issue as discussed in line 185.
>
> Logically, if the model **excessively outputs indicators, its reasoning coherence will be compromised**, naturally leading to lower $R_{ent}$ and $R_{task}$. This will not be encouraged by RL.
>
> >**Re Q3 Stats of Block Size**
>
> We appreciate your suggestion and provide the **b1 detailed stats as follows compared with fixed-size baselines** (baseline with max token length 256, number of block 8 and block size 32):
>
> |Dataset|#Test Data|Avg. #Block|Size Mean|Size Std|Block Size Bin Interval|[1,10]|[11,20]|[21,30]|[31,50]|[51,+∞)|
> |-|-|-|-|-|-|-|-|-|-|-|
> |Countdown|256|10.61|18.50|6.15| |0.26%|84.68%|5.67%|9.28%|0.11%|
> |GSM8k|1319|4.59|31.65|10.39| |0.40%|19.16%|28.65%|46.78%|5.01%|
> |MATH|500|5.78|35.59|12.47| |0.45%|15.19%|31.34%|39.26%|13.77%|
> |KodCode|500|9.47|28.40|9.85| |0.35%|24.00%|35.50%|34.50%|5.65%|
>
> \# denotes numbers. Avg. denotes average. Size Mean and Size Std denote the mean and standard deviation of block size. [1,10] etc., denote the bin size intervals for block size.
>
> The results reflect that **b1 can generate blocks with varied length and adapt to highly variable reasoning step lengths across domains and samples**, where fixed-size blocks cannot handle well. We have added it to the manuscript.
>
> **We sincerely appreciate your highly professional suggestions, which are very helpful in improving our manuscript.** We would be most grateful if you might consider raising the score.

---

> > ### Author Rebuttal · Reviewer_6XN7 · 2026-04-01
> >
> > The rebuttal addresses my main concerns well. The added sensitivity analysis, reward ablations, broader-domain evaluations, and block-size statistics substantially strengthen the paper. My concern about potential segmentation or reward-hacking behavior is partially resolved, but overall the rebuttal increases my confidence in the method. I will therefore raise my score. Thank you for the detailed clarifications and additional experiments, and good luck.

---

> > > ### Author Response · Authors · 2026-04-02
> > >
> > > **We wholeheartedly appreciate your insightful review and recognition of our work.**
> > >
> > > **Given that there is still room to further improve the quality of our work** on the mentioned **reward hacking and potential segmentation**, we conduct further case studies and sincerely would like to hear more from you on these two points.
> > >
> > > > **Reward Hacking without controlling $K$:**
> > >
> > > **Without controlling $K$, we observe clear reward hacking**, where the model **excessively generates indicator tokens and inserts block boundaries**, leading to a severely segmented reasoning chain. This happens because, without the $K$ constraint in the reward, whenever the model generates more indicators, the achieved reward will be higher.
> > >
> > > > **Reward Hacking under Large $K_{target}$:**
> > >
> > > **When $K_{target}$ is extremely large, we also observe an excessive indicator generation that leads to over-segmentation of reasoning coherence.** Different from the situation of not controlling $K$, the model here will try to output more indicators to achieve a higher reward until a saturation at $K_{target}$. Yet due to the constraint of other rewards, the model generally will not reach the $K_{target}$ (too many indicators will impact the reasoning, so that the final answer would be wrong with a low accuracy reward).
> > >
> > >
> > > >**Case Study: Model behaviour for a same reasoning task from GSM8K. The correct answer is 125.**
> > >
> > > | **b1 ($K_{target}$=10)** | **Reward Hacking ($K_{target}$=50)** | **Reward Hacking (w/o $K_{target}$)** |
> > > |------------------------------------------|----------------------------------------|------------------------------------------|
> > > | Step 1: The current value of the jewelry is $5,000. \block | Step 1: The current value of the jewelry is \block\block $5,000. \block\block | Step 1: The current value \block of the jewelry\block\block5000 \block\block |
> > > | Step 2: The jewelry market predicts the jewelry will increase by 2.5%, so the new value will be \(5000 \times 1.025 = 5125\). \block | Step 2: The jewelry market predicts an increase of 2.5%, so compute \block\block 5000 \times 1.025 \block\block which gives 5125. \block\block | Step 2\block\block \block 5000 \times \block 1.02\block\block 5125 \block |
> > > | Step 3: The current value of the electronic gadgets is $8,000. \block | Step 3: The profit from jewelry would be \block\block 5125 \block\block − 5000 = \block\block 125. \block\block | Step 3\block the profit is \block 5125 - 5000 = 125 \block |
> > > | Step 4: The electronic gadgets market predicts the gadgets will increase by 1.2%, so the new value will be \(8000 \times 0.012 = 96\). \block | Step 4: The current value of the electronic gadgets is \block\block $8,000. \block\block | Step\block\block current value \block\block 8000 |
> > > | Step 5: The profit if the merchant buys the jewelry is \(5125 - 5000 = 125\). \block | Step 5: The predicted increase is 1.2%, so calculate \block\block 8000 \times 0.012 \block\block = 96. \block\block | Step\block\block 8000 \times 0.012 \block 96 \block\block |
> > > | Step 6: The profit if the merchant buys the electronic gadgets is \(96\). \block | Step 6: Add this to the original value to get \block\block 8000 + 96 \block\block = 8096. \block\block | Step\block\block 8000 \block 96 = 80\block\block |
> > > | Step 7: To maximise profit, the merchant should choose the jewelry, which yields a profit of $125. \block | Step 7: The profit from electronic gadgets is \block\block 8096 − 8000\block\block= 96. \block\block | Step\block\block answer \block\block jewelry\block\block |
> > >
> > > * The **left column shows that b1** produces well-aligned block segmentation, where each block corresponds to a coherent reasoning step, avoiding disrupting the logic flow and yielding the correct reasoning answer.
> > >
> > > * The **middle column (K = 50) demonstrates reward hacking with extremely large $K_{target}$**, where excessive `\block` tokens fragment reasoning steps and disrupt logical flow, thereby degrading the performance. This observation is consistent with our hyperparameter sensitivity analysis in response to your W1 above.
> > >
> > > * The **right column (w/o $K_{target}$) illustrates a clearer reward hacking behaviour**, where block indicator generation dominates, leading to not only duplicated indicator tokens but broken computations and loss of meaningful reasoning.
> > >
> > > Generally, the results showcase that **(i) unnatural or misaligned block segmentation degrades reasoning coherence, and (ii) without constraints on the generation of block indicators, the model collapses into reward hacking behaviour**.
> > >
> > > **We thank you again for your highly insightful and incisive feedback, which has significantly strengthened our work. We would greatly appreciate your support during the upcoming AC discussion phase.**
> > >
> > > Sincerely,
> > >
> > > Authors

---

### Official Review · Reviewer_92Ud · 2026-03-10

**Soundness:** 4
**Presentation:** 3
**Significance:** 3
**Originality:** 3
**Overall Recommendation:** 5
**Confidence:** 4

**Summary:**

This paper studies the block generation mechanism of diffusion large language models (dLLMs) for reasoning tasks and argues that fixed-size blocks may not align with natural reasoning-step boundaries, which can hurt reasoning coherence and performance. Motivated by empirical observations of block-level entropy patterns in correct versus incorrect reasoning traces, the authors propose b1, a reinforcement learning-based post-training method for learning dynamic-sized reasoning blocks. Specifically, the method introduces a block-ending token to adaptively determine block boundaries and incorporates a Monotonic Entropy Descent (MED) objective to encourage progressively decreasing entropy across adjacent reasoning blocks. The approach is designed as a plug-and-play component that can be combined with existing RL methods for dLLMs. Experiments on GSM8K, MATH500, Sudoku, and Countdown show consistent improvements over fixed-block baselines.

**Compliance With Llm Reviewing Policy:**

Affirmed.

**Final Justification:**

My concerns have been addressed by the author's rebuttal, therefore I am raising my score.

**Key Questions For Authors:**

NA

**Limitations:**

yes

**Strengths And Weaknesses:**

Strengths
1. The dynamic block mechanism is intuitively better aligned with natural reasoning-step boundaries, and the MED objective provides a targeted signal for improving reasoning coherence.
2. b1 is designed as a plug-and-play framework that can be integrated with multiple existing RL methods for dLLMs, which increases its generality and practical value.
3. The paper evaluates the method on multiple reasoning benchmarks and shows consistent improvements over fixed-block baselines.

Weaknesses
1. The paper would benefit from reporting block size statistics after applying b1 (e.g., distribution, mean, or variance), as this would enable a more systematic analysis of the relationship between block length and model performance.

---

> ### Author Rebuttal · Authors · 2026-03-27
>
> **We sincerely appreciate your insightful question, which has greatly helped us improve our paper. We have carefully addressed your concern as follows:**
>
> >**Re W1: Block Stats of b1**
>
> Thank you for the valuable suggestion. We provide the **detailed stats as follows compared with fixed-size baselines** (all fixed-size baseline with max token length 256, number of block 8 and block size 32):
>
> |Dataset|#Test Data|Avg. #Block|Size Mean|Size Std|Block Size Bin Interval|[1,10]|[11,20]|[21,30]|[31,50]|[51,+∞)|
> |-|-|-|-|-|-|-|-|-|-|-|
> |Countdown|256|10.61|18.50|6.15| |0.26%|84.68%|5.67%|9.28%|0.11%|
> |GSM8k|1319|4.59|31.65|10.39| |0.40%|19.16%|28.65%|46.78%|5.01%|
> |MATH|500|5.78|35.59|12.47| |0.45%|15.19%|31.34%|39.26%|13.77%|
> |KodCode|500|9.47|28.40|9.85| |0.35%|24.00%|35.50%|34.50%|5.65%|
>
> \# denotes numbers. Avg. denotes average. Size Mean and Size Std denote the mean and standard deviation of block size (number of tokens within a block). [1,10], [11,20], [21,30], [31,50], and [51,+∞) denote the bin size intervals of histogram distributions for block size.
>
> The results reflect that **b1 can effectively generate blocks with varied length and adapt to highly variable reasoning step lengths for different test samples and domains**, where other fixed-size block baselines cannot achieve that. We have added this analysis to the manuscript.
>
> **We sincerely appreciate your valuable suggestion, which has helped us enrich our manuscript.** We would be glad to address any further questions you may have, and would be most grateful if you might consider raising the score.

---

> > ### Author Rebuttal · Reviewer_92Ud · 2026-04-05
> >
> > The author has fully addressed my confusion, and I will raise my score.

---

> > > ### Author Response · Authors · 2026-04-05
> > >
> > > **We sincerely appreciate your thoughtful review and the constructive perspective provided in your comments, which is highly valuable for improving the manuscript.**
> > >
> > > We would be most grateful for your kind support during the upcoming AC discussion phase.
> > >
> > > Sincerely,
> > >
> > > Authors

---

### Official Review · Reviewer_8ZpN · 2026-03-13

**Soundness:** 3
**Presentation:** 3
**Significance:** 3
**Originality:** 3
**Overall Recommendation:** 5
**Confidence:** 3

**Summary:**

The paper addresses a key issue: fixed block size in dLLM can disrupt reasoning coherence, and proposes a method to learn dynamic blocks using block ending indicators and monotonic entropy descent.

**Compliance With Llm Reviewing Policy:**

Affirmed.

**Final Justification:**

This paper addresses a critical issue by explicitly introducing dynamic block division into dLLM training via reinforcement learning. It fully resolved my confusion during the rebuttal stage, and I consider it a valuable work worthy of acceptance.

**Key Questions For Authors:**

1. Does Theorem 1 only hold true when all block entropies are distinct and strictly orderable? Does equivalence still hold if there are identical entropies, nearly identical entropies, or different block numbers K? Could you elaborate on the theorem's applicability boundaries?
2. Why is the comparison performed under a 0-shot setting where AdaBlock is clearly inferior? How does b1 perform under a few-shot setting?
3. Could the model obtain Rind's reward hack by frequently outputting \blocks?
4. Are the settings α=β=γ=1 in the reward design merely empirical choices? Is this parameter design robust across tasks?
5. Can the method be generalized to non-mathematical reasoning or non-CoT-heavy tasks, such as code generation or multimodal reasoning tasks?

**Limitations:**

yes, the authors adequately discuss the limitations and potential negative societal impact of their work.

**Strengths And Weaknesses:**

Strengths: This paper focuses on a key issue, explicitly introducing dynamic block division into dLLM training using RL, and attempting to use Monotonic Entropy Descent as a training signal for reasoning coherence. The paper provides ample experimental evidence to support this result.


Weaknesses: Some conclusions in this paper still require further experimental verification and theoretical analysis. For details, please refer to Key Questions For Authors.

---

> ### Author Rebuttal · Authors · 2026-03-27
>
> **We sincerely thank your insightful concerns, and we have thoroughly addressed your concerns as follows:**
>
> > **Re Q1.1: Theorem 1 Applicability Boundaries**
>
> Theorem 1 holds across diverse boundary conditions because it establishes an equivalence of the optimal structure (argmax) rather than absolute values (max) between the global rank-based objective and its local pairwise reward signal. While the maximum numerical scores of $R_{ent}$ and $r_{SCC}$ may vary, the equivalence lies in their shared preference for monotonic orderings, yielding consistent optima.
>
> **Identical entropies:** While maximum values of the two objectives may differ, the set of optimal solutions remains aligned, as tied elements contribute equally to both objectives. Therefore, the equivalence still holds at the level of optimisation (argmax).
>
> **Nearly identical entropies:** Although rankings may be sensitive to small perturbations, this is inherent to rank-based metrics, and does not affect the equivalence in Theorem 1, as both objectives depend only on relative ordering.
>
> **Different K:** Theorem 1 is independent of the block number since $K$ only changes the size of the permutation space, not the ordering structure.
>
> > **Re Q1.2: Extension for Tie Cases**
>
> To elevate the theoretical rigour and ensure that both the maximum value (max) and optimal structure (argmax) align under ties, we have extended Theorem 1 using a tie-aware Spearman definition: $r_{SCC} = -\rho_S(\mathcal{H}(\mathbf{b}_1^d), \dots, \mathcal{H}(\mathbf{b}_K^d))$
>
> Here $\rho_S$ is the rank correlation coefficient computed by average ranks for tie handling. Under this definition, an identical entropy sequence yields exactly zero for the global $r_{SCC}$. This rigorously matches our local reward, where $R_{ent} = 0$ for ties. This modification ensures both metrics penalise ties equally and also preserve the same optimisation equivalence in Theorem 1. **We appreciate your help and have updated our manuscript with this formal extension.**
>
> >**Re Q2: 0-shot and Few-shot**
>
> We **follow baselines** d1 and wd1 to use a 0-shot setting for **fair and consistent comparisons**.
>
> Following your suggestion, we conduct **additional few-shot experiments** on 5-shot (GSM8K) and 4-shot (MATH) following AdaBlock:
>
> |   | LLaDA | AdaBlock | wd1  | b1   |
> |----------|----------|----------|------|------|
> | GSM8K, 5-shot |    76.7      |     78.5     |   78.4   |   81.2   |
> | MATH, 4-shot  |      35.2    |     35.8     |    37.2  |    37.8  |
>
> Our **b1 achieves consistent improvement** over baselines in **few-shot setting**.
>
> >**Re Q3: $R_{ind}$ Reward Hack**
>
> Thank you for the insightful question. Our design of $R_{ind}$ in Eq. 5 ensures that **if too many blocks are generated, the reward will be capped at 1**. This is controlled by a threshold $K_{target}$ (Details on $K$ can be found in response to Reviewer 6XN7 W1 due to word limit). In this way, b1 prevents the potential indicator reward hacking issue as discussed in line 185.
>
> Logically, if the model **frequently outputs indicators, its reasoning coherence will be compromised**, naturally leading to lower $R_{ent}$ and $R_{task}$. This will not be encouraged by RL.
>
> >**Re Q4: $\alpha=\beta=\gamma=1$**
>
> As discussed in line 221, we **follow baselines**, d1, wd1 and GDPO, to set all reward weights to 1 to **ensure fair comparison**.
>
> We provide further **sensitivity analysis** on HumanEval by changing one weight while fixing the other two. The results verify **b1's robustness**.
>
> | | **$\alpha$** | **$\beta$** | **$\gamma$** |
> | :--- | ---: | ---: | ---: |
> | 0.0 | 30.49 | 31.10 | 29.88 |
> | 0.3 | 31.10 | 31.71 | 30.49 |
> | 0.5 | 31.71 | 32.32 | 31.71 |
> | 0.7 | 32.32 | 32.93 | 31.71 |
> | 1.0 | 32.32 | 32.32 | 32.32 |
>
> >**Re Q5: Other Tasks**
>
> Following your suggestions, we further evaluate b1 on **code generation** (MBPP, HumanEval and KodCode), **world knowledge** (MMLU and MMLU-Pro) and **scientific comprehension** (ARC-E and ARC-C) tasks.
>
> |Method|HumanEval|MBPP|KodCode|MMLU|MMLU-Pro|ARC-E|ARC-C|
> |:---|:---|:---|:---|:---|:---|:---|:---|
> |LLaDA|28.66|33.60|24.00|53.46|33.79|76.98|73.21|
> |d1|29.88|34.40|25.20|55.56|34.68|77.23|75.60|
> |**d1+b1**|**31.10**|**35.00**|**27.00**|**58.98**|**36.32**|**80.93**|**78.67**|
> |**${\Delta}$**|**+1.22**|**+0.60**|**+1.80**|**+3.42**|**+1.64**|**+3.70**|**+3.07**|
> |wd1|30.49|34.00|25.60|57.77|35.84|81.14|78.50|
> |**wd1+b1**|**32.32**|**35.80**|**29.00**|**62.22**|**38.21**|**90.53**|**81.48**|
> |**${\Delta}$**|**+1.83**|**+1.80**|**+3.40**|**+4.45**|**+2.37**|**+9.39**|**+2.98**|
>
> Results show that **b1 exhibits strong generalisability across various non-mathematical reasoning and non-CoT reasoning** tasks. Multimodal reasoning is a promising direction yet possibly out of the scope of this paper. We would love to explore in future work.
>
> **We sincerely appreciate your valuable feedback, which greatly helped us improve the manuscript.** We would be most grateful if you might consider raising the score.

---

> > ### Author Rebuttal · Reviewer_8ZpN · 2026-04-03
> >
> > The author has fully addressed my confusion, and I will raise my score.

---

> > > ### Author Response · Authors · 2026-04-03
> > >
> > > **We wholeheartedly appreciate your insightful review and recognition of our work. Your comments provide a clear and valuable perspective that greatly helps improve our paper.**
> > >
> > > We would be most grateful for your support during the upcoming AC discussion phase.
> > >
> > > Sincerely,
> > >
> > > Authors

---

### Official Review · Reviewer_Vtq3 · 2026-03-13

**Soundness:** 3
**Presentation:** 3
**Significance:** 3
**Originality:** 4
**Overall Recommendation:** 5
**Confidence:** 4

**Summary:**

This paper studies a limitation of block-based diffusion LLMs: fixed-size blocks can misalign with reasoning-step boundaries and the best block size varies across tasks. The authors propose b1, which learns dynamic block boundaries via an indicator token and RL, and adds two rewards: an indicator reward for sufficient reasoning steps and a Monotonic Entropy Descent (MED) reward that encourages decreasing block entropy across blocks. Experiments on LLaDA-8B-Instruct over GSM8K, MATH500, Countdown, and Sudoku (trained with Diffu-GRPO, GDPO, d1, and wd1) show consistent gains over fixed-size baselines, especially on Countdown.

**Compliance With Llm Reviewing Policy:**

Affirmed.

**Final Justification:**

My concerns have been addressed by the author's rebuttal, therefore I am raising my score.

**Key Questions For Authors:**

My questions follow directly from the weaknesses.

Additionally:
1. The paper argues that optimal fixed block size varies by task, what is the block size used for the fix-block-size methods test in Table 1?

**Limitations:**

yes.

**Strengths And Weaknesses:**

## Strengths
1. Well-motivated problem. The entropy visualizations make the failure mode of fixed blocks intuitive and convincing.
2. Simple, modular method. The indicator token + indicator reward + MED reward is easy to understand and easy to plug into existing dLLM RL pipelines.
3. Consistent empirical gains. Improvements appear across tasks and RL baselines, and the ablations suggest both rewards contribute.

## Weaknesses
1. MED is validated as a useful signal, but not fully isolated as the key mechanism. The paper shows ablation and correlation evidence, but it does not compare against other intrinsic reward such as directly encouraging low average block entropy, or even semantic rule based rewards (i.e. adding /block after each "step").
2. Would have more generality by testing on wider range of task domains. Evaluation is on the classic set of math/logical reasoning tasks (solution is mostly stepwise), so it is unclear how broadly the method transfers to tasks where the solutions' segmentation signals are weak.
3. The argument for efficiency would be strengthened by showing stats on the length of generated tokens and number of generated blocks. High throughput might be a result of shorter inference steps (inspired by Figure 7).

---

> ### Author Rebuttal · Authors · 2026-03-27
>
> **We sincerely thank you for your valuable question**, and we have thoroughly addressed all your concerns. We summarise our response and **highlight** the key information as follows:
>
> >**Re W1: MED & Other Rewards**
>
> We appreciate this insightful suggestion. We conducted further ablation. Specifically, **3 more intrinsic rewards**: (i) a **low-block entropy reward** (to encourage low averaged block entropy), (ii) a **format reward** (valid block count), and (iii) a **verbosity reward** (block length penalty) are compared with our block entropy reward.
>
> |  | MATH | GSM8K | HumanEval | MBPP |
> | :--- | :---: | :---: | :---: | :---: |
> | **wd1 (fixed-size)** | 34.2 | 78.9 | 30.5 | 34.0 |
> | **b1 (MED reward)** | **37.4** | **80.8** | **32.3** | **35.8** |
> | **b1 (avg block entropy reward)** | 34.6 | 79.3 | 29.9 | 34.4 |
> | **b1 (format reward)** | 33.8 | 78.2 | 28.7 | 33.6 |
> | **b1 (block size reward)** | 34.4 | 79.4 | 30.5 | 33.2 |
>
>
> The results show that **MED-based block entropy reward gives the main improvement** compared to other intrinsic rewards, such as the average block entropy reward, which directly encourages low average block entropy. This further isolates MED as the key mechanism for b1's effectiveness. We have added this to our manuscript.
>
> >**Re W2: Generalisation to Other Domains**
>
> We further evaluate b1 on **code generation** (MBPP, HumanEval and KodCode), **world knowledge** (MMLU and MMLU-Pro) and **scientific comprehension** (ARC-E and ARC-C) tasks.
>
> |Method|HumanEval|MBPP|KodCode|MMLU|MMLU-Pro|ARC-E|ARC-C|
> |:---|:---|:---|:---|:---|:---|:---|:---|
> |LLaDA|28.66|33.60|24.00|53.46|33.79|76.98|73.21|
> |d1|29.88|34.40|25.20|55.56|34.68|77.23|75.60|
> |**d1+b1**|**31.10**|**35.00**|**27.00**|**58.98**|**36.32**|**80.93**|**78.67**|
> |**${\Delta}$**|**+1.22**|**+0.60**|**+1.80**|**+3.42**|**+1.64**|**+3.70**|**+3.07**|
> |wd1|30.49|34.00|25.60|57.77|35.84|81.14|78.50|
> |**wd1+b1**|**32.32**|**35.80**|**29.00**|**62.22**|**38.21**|**90.53**|**81.48**|
> |**${\Delta}$**|**+1.83**|**+1.80**|**+3.40**|**+4.45**|**+2.37**|**+9.39**|**+2.98**|
>
> For the additional experiments, the default block size for the fixed-size baseline is 32, and the generation length is 256. For code generation, within the Python tag, we delete the \block tag after generation to avoid code syntax errors for testing. For world knowledge and comprehension tasks, we adopt the same \<reasoning\> then \<answer\> format template as used in d1. Note that MBPP and HumanEval do not provide training splits. Therefore, we follow GDPO and use the KodCodeLight-RL-10K [1] benchmark as the default GRPO training dataset for all code generation tasks and randomly split 500 samples for the test set of KodCode.
>
> The results show that **b1 consistently outperforms** the fixed-size baselines on tasks **from additional domains**, verifying the generalisability across domains with different solution segmentation patterns. We appreciate your suggestions, and we have added this to our manuscript.
>
> >**Re W3: Statistics of Generated Blocks**
>
> Thank you for this valuable suggestion. We provide the **detailed stats as follows compared with fixed-size baselines** (max token length 256, number of block 8 and block size 32):
>
> |Dataset|#Test Data|Avg. #Block|Size Mean|Size Std|Block Size Bin Interval|[1,10]|[11,20]|[21,30]|[31,50]|[51,+∞)|
> |-|-|-|-|-|-|-|-|-|-|-|
> |Countdown|256|10.61|18.50|6.15| |0.26%|84.68%|5.67%|9.28%|0.11%|
> |GSM8k|1319|4.59|31.65|10.39| |0.40%|19.16%|28.65%|46.78%|5.01%|
> |MATH|500|5.78|35.59|12.47| |0.45%|15.19%|31.34%|39.26%|13.77%|
> |KodCode|500|9.47|28.40|9.85| |0.35%|24.00%|35.50%|34.50%|5.65%|
>
> \# denotes numbers. Avg. denotes average. Size Mean and Size Std denote the mean and standard deviation of block size (number of tokens within a block). [1,10], [11,20], [21,30], [31,50], and [51,+∞) denote the bin size intervals of histogram distributions for block size.
>
> The results reflect that b1 can effectively generate blocks **with varied length and adapt to highly variable reasoning step lengths across test samples and domains**, where fixed-size blocks can not achieve it. We have added this analysis to the manuscript.
>
> We also kindly clarify that throughput is independent of reasoning steps, as it measures decoded tokens per second rather than step count. b1 can generate longer/shorter blocks compared with fixed-size baselines. Therefore, we are not strongly aiming to claim significant efficiency improvement.
>
> >**Re Q1 Block Size for Baselines in Table 1**
>
> For all base methods in Table 1, the default block size is 32. More details of the default hyperparameters are in Table 5.
>
> **We thank you again for your constructive feedback, which helps us improve the quality of this manuscript.** If you have any further questions, we would be more than happy to discuss them with you. We would sincerely appreciate it if you might consider raising the score.
>
> [1] Xu et al., KodCode: A Diverse, Challenging, and Verifiable Synthetic Dataset for Coding. ACL 2025.

---

> > ### Author Rebuttal · Reviewer_Vtq3 · 2026-04-04
> >
> > The authors have addressed my concerns. I will raise my scores. Good luck.

---

> > > ### Author Response · Authors · 2026-04-04
> > >
> > > **We wholeheartedly appreciate your thoughtful review and recognition of our work. Your comments offer a clear and constructive perspective that helps improve the paper.**
> > >
> > > We would greatly appreciate your support during the upcoming AC discussion phase.
> > >
> > > Sincerely,
> > >
> > > Authors

---

### Decision · Program_Chairs · 2026-04-30

**Decision:**

Accept (regular)

**Comment:**

This submission proposes b1, a reinforcement learning framework designed to learn dynamic-size reasoning blocks for diffusion large language models (dLLMs) through a monotonic entropy descent objective. All four reviewers recommend acceptance, praising the method's motivation, its plug-and-play modularity, and its consistent performance improvements over fixed-block baselines. The authors provided a comprehensive rebuttal that addressed initial questions concerning task generalizability, reward-hacking behaviors, and theoretical applicability boundaries. Specifically, additional experiments on coding and world knowledge benchmarks, along with detailed block-size statistics, effectively demonstrated the framework's versatility and effectiveness. Having monitored the discussion and reviewed the author responses, I concur with the reviewer consensus that the work is technically sound and provides a valuable solution to a known bottleneck in dLLM decoding. The paper is well-suited for the ICML community.